# Enabling Your Forensic Detector Know How Well It Performs on Distorted Samples

**Bin Li**[1], **Haoyu Li**[1], **Haodong Li**[1,*], **Jiaming Zhong**[1]
**Changsheng Chen**[2], **Jiangqun Ni**[3], **Bo Cao**[4]

[1]Guangdong Provincial Key Laboratory of Intelligent Information Processing and
Shenzhen Key Laboratory of Media Security, Shenzhen University
[2]Faculty of Engineering, Shenzhen MSU-BIT University [3]SUN YAT-SEN University
[4]The Smart City Research Institute of China Electronics Technology Group Corporation

## ABSTRACT

Generative AI has substantially facilitated realistic image synthesizing, posing great challenges for reliable forensics. When image forensic detectors are deployed in the wild, the inputs usually undergone various distortions including compression, rescaling, and lossy transmission. Such distortions severely erode forensic traces and make a detector fail silently—returning an over-confident binary prediction while being incapable of making reliable decision, as the detector cannot explicitly perceive the degree of data distortion. This paper argues that reliable forensics must therefore move beyond "is the image real or fake?" to also ask "how trustworthy is the detector's decision on the image?" We formulate this requirement as Detector's *Distortion-Aware Confidence* (**DAC**): a sample-level confidence that a given detector could properly handle the input. Taking AI-generated image detection as an example, we empirically discover that detection accuracy drops almost monotonically with full-reference image quality scores as distortion becomes severer, while such references are in fact unavailable at test time. Guided by this observation, the *Distortion-Aware Confidence Model* (**DACOM**) is proposed as a useful assistant to the forensic detector. DACOM utilizes full-reference image quality assessment to provide oracle statistical information that labels the detectability of images for training, and integrates intermediate forensic features of the detector, no-reference image quality descriptors and distortion-type cues to estimate DAC. With the estimated confidence score, it is possible to conduct selective abstention and multi-detector routing to improve the overall accuracy of a detection system. Extensive experiments have demonstrated the effectiveness of our approach.

## 1 INTRODUCTION

With the rapid development of generative artificial intelligence (*e.g.*, GANs (Goodfellow et al., 2014) and Diffusion Models (Ho et al., 2020)), photo-realistic contents can be manipulated or synthesized at scale. To defense against such fake visual contents, image forensics increasingly underpin safety-critical decisions. However, great challenges are presented in the practical deployment of forensic detectors. Most images have already undergone various degradations, such as compression, resampling, lossy platform transmission, and so on. Such distortions which may not be well-informed can weaken forensic traces and often push an input beyond the detector's "comfort zone". Unfortunately, if a detector is trained exclusively on clean data, it remains largely unaware of distortions; consequently, as shown in Figure 1 (a), it reports only a binary decision (real or fake) without quantifying how much the decision should be trusted. Lacking this confidence signal, downstream systems (*e.g.*, human fact-checkers, multi-detector pools) cannot assess which detector performs better on a certain sample, and thus are unable to reasonably abstain or select among detectors, raising both reliability and usability concerns. Even when distortion augmentations are included during training, this issue still persists. Consequently, we argue that confidence matters in reliable image forensics.

---

*Corresponding Author

To tackle the above challenges, several intuitive strategies could be employed in practice, yet they suffer from clear limitations. Robustness training seeks a model that works under distortions, but the combinatorial explosion of distortion type/degree makes it elusive and computationally costly. Confidence calibration (Guo et al., 2017) can improve probability estimates in standard classification, but distortion shifts the input distribution heterogeneously and thus breaks the calibration assumptions. Using image quality (assessed without reference) as a proxy for distortion level is potential to estimate forensic performance, as shown in prior study (Kim et al., 2024). However, common no-reference image quality assessment (NR-IQA) is designed for human perception and cannot directly reflect the detection confidence (refer to Section 3). Another line of work examines forensicability (Chu et al., 2015; Pasquini & Böhme, 2018)—the intrinsic detectability determined by the data distribution. However, such treatments do not bridge detectability of data with specific detector, leaving a gap in practical application. Therefore, we propose constructing a *Distortion-Aware Confidence* (**DAC**) score, a detector-conditioned score that estimates the probability that a given image will be classified correctly under its present distortions.

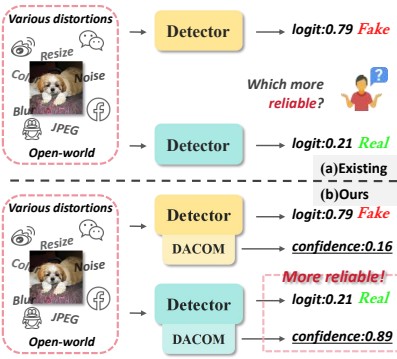

Figure 1: When facing open-world distortions, detectors' outputs lack reliability. The proposed Distortion-Aware Confidence Model (DACOM) provides sample-level confidence scores to facilitate reliable detection.

In this paper, we take AI-generated image (AIGI) detection as a case study to analyze how distortion affects forensic performance and how to link image quality with DAC. Our analysis reveals that, when a pristine reference is available, full-reference image quality assessment (FR-IQA) produces scores that line up almost monotonically with detector accuracy across all tested distortions, suggesting that forensic confidence can be obtained with the guidance of image quality. The gaps for practical applications lie in: (i) Image quality must be estimated without reference at test time. While NR-IQA is available, it is too noisy to serve as a direct surrogate. (ii) Our analysis shows that different types of distortions lead to different degrees of detection confidence, even when the resulting FR-IQA score is the same. It indicates that we need to build a more sophisticated and generalizable model to bridge image quality, distortion type, and forensic performance, so as to effectively estimate DAC which can be served as an useful indicator for either human decision or expert system fusion.

Building on the above insights, this paper proposes to develop a *Distortion-Aware Confidence Model* (**DACOM**). The key idea is to distill the reliable relationship observed with FR-IQA into a trainable confidence model, eliminating the need for references at test time. DACOM fuses forensic features derived from the detector, no-reference image quality descriptors, and distortion-type cues to produce confidence scores. During training, for each distortion type, we record the detector's empirical accuracy at different levels of image quality by using FR-IQA as an oracle, and treat the corresponding accuracy as image label explicitly linking distortion degree to forensic performance. A regressor is trained to map the features to the obtained labels. Once trained, DACOM outputs a confidence score per image (Figure 1 (b)), which can be attached to legacy detectors or to an ensemble of detectors, enabling abstention policies and multi-detector routing. Our contributions are summarized as follows:

- We formulate DAC, providing a principled reliability target for practical forensics. On this basis, we conduct an in-depth analysis and reveal the relationship between FR-IQA scores and forensic accuracy, offering the empirical foundation for confidence modelling.

- We introduce DACOM to connect image quality with detection confidence. It uses supervision guided by FR-IQA during training, yet at inference it estimates sample-level DAC scores without reference. DACOM can be deployed with diverse forensic detectors.

- Extensive experiments on multiple datasets, detectors, and distortion types shows that DACOM can effectively predict confidence and unlock advantages in downstream tasks, including a 7.66% relative accuracy improvement via selective filtering and a 5.84% accuracy improvement in multi-detector routing compared with naive logit calibration.

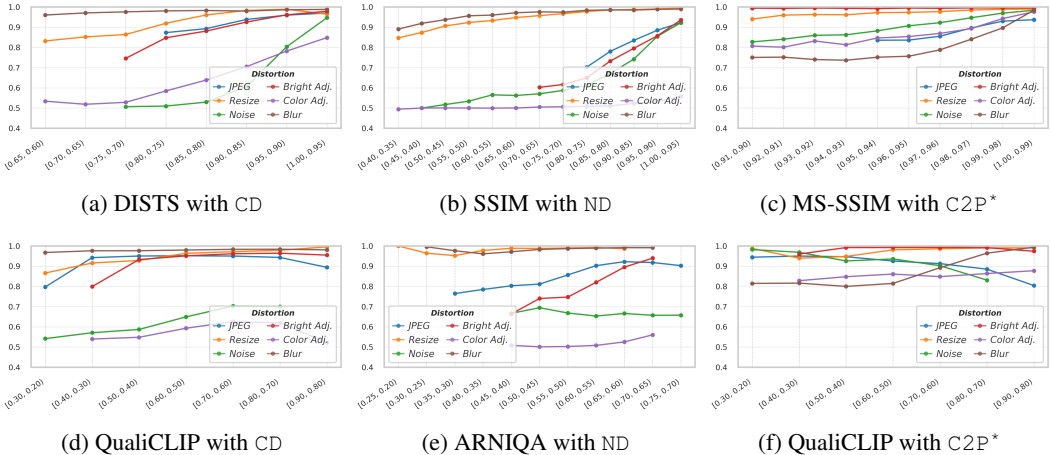

(a) DISTS with `CD`     (b) SSIM with `ND`     (c) MS-SSIM with `C2P*`

(d) QualiCLIP with `CD`     (e) ARNIQA with `ND`     (f) QualiCLIP with `C2P*`

Figure 2: Relationship between IQA Metrics and Model Detection Performance. (a)-(c) show the relationship between FR-IQA scores and model balanced accuracy; (d)-(f) show the corresponding relationship for NR-IQA metrics.

## 2 RELATED WORKS

**Robust AIGI Detection.** Early CNN-based detectors focused on spatial features and employed data augmentation (*e.g.*, JPEG compression, blurring) (Wang et al., 2020; Gragnaniello et al., 2021) to gain robustness against common distortions, but they generalized poorly to unseen processing. Later methods exploited frequency features (Tan et al., 2024a; 2023; Li et al., 2024b; Tan et al., 2024b), handcrafted extractors (Li et al., 2024a) to enhance subtle forensic cues and improve generalization. Yet, frequency-domain information is highly distortion-sensitive, where even slight perturbations can cause sharp performance drops. Some approaches (Tao et al., 2025) improved robustness to JPEG and OSN (Online Social Networks) distortions by adding paired original-compressed samples during training. More recently, reconstruction-based methods use diffusion models to recover authentic counterparts, where latent-space reconstruction (Chen et al., 2024a) yields greater robustness than pixel-space (Wang et al., 2023) under the same augmentations. While these methods improve robustness, they seldom address the reliability of predictions under diverse distortions. In contrast, our work focuses on quantifying reliability by linking distortion intensity to detection performance.

**Model Self-Assessment and Forensibility.** Many methods have been proposed for uncertainty estimation (Blundell et al., 2015; Gal & Ghahramani, 2016; Lakshminarayanan et al., 2017; Van Amersfoort et al., 2021; 2020; Guo et al., 2017; Liang et al., 2017; Naeini et al., 2015), but they are mostly designed for general tasks and have not been systematically adapted to image forensics. Recently, several studies (Zhang et al., 2025; Ji et al., 2023; Lin et al., 2024; Hao et al., 2024; Pan et al., 2024) have introduced uncertainty modeling into forensics, yet typically as a means to boost model performance or robustness, rather than as a principled quantification of detection reliability. In addition, the information-theoretic notion of forensicability (Chu et al., 2015; Pasquini & Böhme, 2018; Chen et al., 2022; Pasquini & Böhme, 2017; Schlögl et al., 2021; Li et al., 2023) provides valuable insights into the theoretical limits of forensic systems. However, its application has largely remained at a theoretical level, with limited validation under realistic conditions, leading to a gap between theoretical potential and practical effectiveness.

## 3 ANALYSIS OF FORENSIC PERFORMANCE: A DISTORTION PERSPECTIVE

Before introducing our confidence model, we first present an analysis of forensic performance from a distortion perspective, establishing the empirical foundation for our framework. Specifically, we aim to answer two questions:

(i) **How do common distortions affect the accuracy of forensic detectors?**

(ii) **Can off-the-shelf Image Quality Assessment (IQA) metrics serve as proxies for estimating the detection confidence on a given image?**

**Setup of Analysis.** We adopt three pretrained AIGI detectors (CD, ND, and C2P⋆; details provided in Section 5.1) and six common distortion types (additional results for more distortion types is provided in Appendix Figure 6) in our empirical analysis. For every distorted image, we employ image quality assessment (IQA) methods to quantify the degree of distortion. Specifically, we compute FR-IQA scores—DISTS (Ding et al., 2020), SSIM (Hore & Ziou, 2010), and MS-SSIM (Sara et al., 2019)—using the pristine reference, as well as NR-IQA scores—ARNIQA (Agnolucci et al., 2024b), QualiCLIP (Agnolucci et al., 2024a), *etc*. We then bucket images by quality score and measure balanced accuracy in each bucket, thereby obtaining the forensic performance of each detector on images subjected to a certain type and a certain degree of distortion. The obtained results are shown in Figure 2.

**FR-IQA strongly correlates with detection performance.** A clear positive correlation between FR-IQA metrics and the performance of different detectors is observed in Figure 2 (a)-(c): for a fixed distortion type, a more severe degradation (*i.e.*, a lower FR-IQA score) results in poorer detection performance. This suggests that FR-IQA can reliably characterize the relation between distortion intensity and performance degradation under a given distortion type. Moreover, we observe that the more severe FR-IQA degradation, the greater shifts in the logit distribution (Appendix B). Therefore, we conclude that *FR-IQA captures an oracle notion of distortion severity that is strongly predictive of forensic performance*. We should also note that *FR-IQA methods cannot be applied at inference time as they require access to the reference images*.

**NR-IQA alone is insufficient to predict detection performance.** In contrast to FR-IQA, the correlation between NR-IQA metrics and detection performance is relatively weak. As shown in Figure 2 (d)-(f), the NR-IQA scores fail to exhibit a consistent trend of "lower quality leads to lower performance". This is primarily because NR-IQA metrics are designed based on human perceptual quality, which is highly content-dependent and thus difficult to be directly applied to predicting the behavior of forensic models. These observations indicate that *while NR-IQA methods are deployable and can perceive degradation, their scores exhibit a weak and inconsistent correlation with forensic performance, making them unsuitable to be used alone as a reliable indicator of model confidence*.

**Distortion type is a critical factor.** As shown in Figure 2, even at the same level of distortion, different distortion types lead to noticeably different accuracies, no matter FR-IQA or NR-IQA is used. This implies that *distortion type is an indispensable factor for assessing forensic performance.*

The above findings confirm that both the distortion type and its severity determine forensic performance. While FR-IQA provides a dependable signal, it is inherently reference-dependent. At test time, only NR-IQA scores and detector-internal features are available, and either source alone is insufficient to predict detection confidence. These observations motivate a reference-free, distortion-aware confidence model that (i) learns to distill the relationship between FR-IQA scores and detection performance during training, and (ii) fuses detector features with cues of distortion type and severity at inference. The proposed model is described in the next section.

## 4 METHODOLOGY

In this section, we first formalize the goal of *Distortion-Aware Confidence* (DAC), then detail a two-stage pipeline that (i) converts full-reference image quality statistics into per-image reliability labels and (ii) trains a reference-free Distortion-Aware Confidence Model (DACOM) that can be plugged into a given detector.

### 4.1 PROBLEM FORMULATION

Consider an image forensic detector $M : x \in \mathbb{R}^{H \times W \times 3} \to c \in \{0, 1\}$ trained in the standard binary-classification paradigm. A real-world test image $x$ is typically subjected to an (unknown) distortion operation $\phi_{t,s}$—characterised by its type $t \in \mathcal{T}$ and severity $s$—to a pristine image $x_0$: $x = \phi_{t,s}(x_0)$. As distortion attenuates forensic traces, the softmax or logit value of detector $M$ on $x$ is no longer a trustworthy indicator of correctness. Our target thus is to obtain a DAC score, which is defined as the *sample-level probability that $M$ is correct*:

$$\text{DAC}_M(x) = \text{Pr}(M(x) = c \mid x) \in [0, 1]. \tag{1}$$

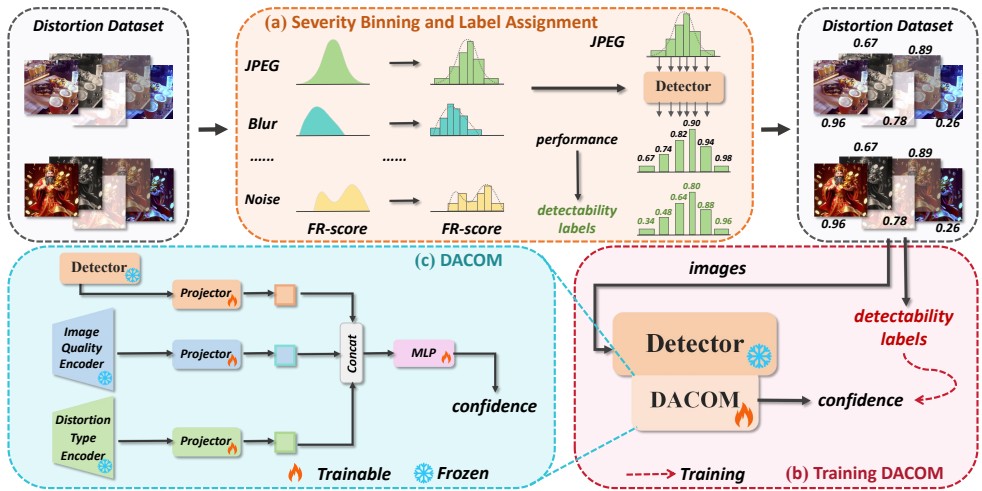

Figure 3: Overview of the distortion-aware confidence pipeline. (a) illustrates the process of adaptive severity binning and data labeling, where prior knowledge of forensic performance is used to assign labels to data within each intensity bin for every distortion type. (b) depicts the training process of the DACOM. (c) shows the architecture of the DACOM.

At test time we assume no access to the pristine reference $x_0$ and the class label $c$; only the image $x$ itself and intermediate features of $M$ are available.

## 4.2 DISTORTION-AWARE CONFIDENCE PIPELINE

As already revealed in Sec. 3, detector accuracy varies *monotonically* with the FR-IQA score, and NR-IQA and detector embeddings are readily available at runtime but neither alone aligns well with detection accuracy. These observations inspire us to adopt a two-stage pipeline as illustrated in Fig. 3 (a)-(b). Stage A — *Collect statistics and perform labeling.* For every distortion type $t$ we use FR-IQA scores $q_{\mathrm{FR}}(x, x_0)$ to build an ordered distortion severity axis, divide it into several bins, bucket image according to bins, and record the detector's balanced accuracy as a label of "detectability prior". Stage B — *Train a reference-free predictor.* We train DACOM, a regressor that fuses three inference-available feature streams, including detector embeddings, NR-IQA descriptors, and distortion-type cues, and regresses to the detectability label obtained in Stage A. Once trained, DACOM outputs $\hat{s}(x; M) \approx \mathrm{DAC}_M(x)$ for a given image without requiring reference images.

## 4.3 STAGE A: SEVERITY BINNING AND LABEL ASSIGNMENT

We construct a large-scale distorted dataset that covers multiple distortion types $t \in \mathcal{T}$ and a range of severities. Guided by FR-IQA, we build, *within each type $t$*, a unified severity axis and bucket the samples along this axis. Since different distortion types exhibit distinct ranges and/or distributions of FR-IQA scores (Appendix C), naive uniform partitioning by absolute FR-IQA values causes severe imbalance across types and bins. To avoid this, we adopt *type-wise adaptive binning* with a fixed number of bins $B$ for every $t$, so that each distortion type contributes $B$ buckets with comparable sample sizes.

**Type-wise Adaptive Binning.** Let $q_{\mathrm{FR}}$ be an FR-IQA metric whose larger value indicates higher quality. For each distortion type $t$ we split its FR-IQA score distribution into $B$ equal-frequency bins and collect samples into each bin as:

$$\mathrm{bin}(t, b) = \{x \mid t(x) = t, q_t^{(b-1)} \leq q_{\mathrm{FR}}(x, x_0) < q_t^{(b)}\}, b = 1 \ldots B, \tag{2}$$

where $t(x)$ is the distortion type of $x$ and $q_t^{(b)}$ are the $\frac{b}{B}$-quantiles w.r.t. distortion type $t$. Quantile binning guarantees similar sample counts per $(t, b)$ and avoids the imbalance caused by the different dynamic ranges of FR-IQA metrics across types. In addition, a dedicated bin is reserved for pristine images, *i.e.*, $\mathrm{bin}(\texttt{pristine}, 1)$.

**Bin-wise Detectability Labeling.** For each bin, we compute the detector's balanced accuracy on samples lying in the bin:

$$\text{BAcc}_{t,b} = \text{Acc}^{\text{bal}}\left(M; \text{bin}(t,b)\right). \tag{3}$$

This bin-level statistic serves as a *statistical average*, which is then converted into a scalar label for representing the degree of a sample is expected to be correctly classified by $M$. Specifically, $\text{BAcc}_{t,b}$ is mapped to a *detectability label* within the range of $[0, 1]$:

$$y_{t,b} = 2 \cdot \max\left(\text{BAcc}_{t,b} - 0.5, 0\right), \tag{4}$$

where $y_{t,b} = 0$ means random guessing and $y_{t,b} = 1$ means perfect detection. Every sample $x \in \text{bin}(t,b)$ inherits the label $y(x) = y_{t,b}$. These labels encode *both* distortion severity and type influence, providing supervised targets for Stage B.

## 4.4 STAGE B: CONSTRUCTING THE DISTORTION-AWARE CONFIDENCE MODEL

As illustrated in Fig. 3 (c), for an image $x$ DACOM first extracts features with three complementary encoders:

- **Forensic Trace Encoder** $\phi_M$: it extracts intermediate features from the frozen forensic detector $M$, carrying information about forensic traces and their corruption.
- **Image Quality Encoder** $\phi_{\text{IQ}}$: it extracts distortion-sensitive features by using an NR-IQA descriptor.
- **Distortion Type Encoder** $\phi_{\text{DT}}$: it extracts embeddings of a distortion-type classifier, which is capable of identifying various distortion types.

Features from each branch are then linearly projected to a $D$-dimensional space ($D$=256):

$$\mathbf{z}_M = \mathbf{W}_M \phi_M(x), \quad \mathbf{z}_{\text{IQ}} = \mathbf{W}_{\text{IQ}} \phi_{\text{IQ}}(x), \quad \mathbf{z}_{\text{DT}} = \mathbf{W}_{\text{DT}} \phi_{\text{DT}}(x). \tag{5}$$

and optionally subjected to LayerNorm. The concatenated feature vector $\mathbf{h}(x) = [\mathbf{z}_M \| \mathbf{z}_{\text{IQ}} \| \mathbf{z}_{\text{DT}}] \in \mathbb{R}^{3D}$ is fed to a MLP head $g_\theta$ for predicting the distortion-aware confidence:

$$\hat{s}(x; M) = g_\theta\left(\mathbf{h}(x)\right) \in [0, 1]. \tag{6}$$

**Training Objective.** Given the bin-derived label $y(x)$, DACOM minimises a weighted mean-squared error:

$$\mathcal{L}_{\text{MSE}} = \frac{1}{N} \sum_{i=1}^{N} w_{t_i, b_i} \left(\hat{s}(x_i; M) - y(x_i)\right)^2, \tag{7}$$

where $w_{t,b}$ inversely scales with the bin's sample size to counter residual imbalance. Empirically, this simple loss can preserve the monotonicity FR-IQA observed in Sec. 3.

**Inference and Usage.** At deployment, DACOM needs only the three features extracted from a given image to output $\hat{s}(x; M)$. The output score enables: (i) *selective abstention*: refrain from making decisions when $\hat{s}(x; M)$ is insufficiently high; and (ii) *multi-detector selection*: given multiple detectors $\{M_j\}$, pick the one with the highest $\hat{s}(x; M_j)$ for each input $x$. Both applications can enhance overall reliability.

## 5 EXPERIMENTS

### 5.1 EXPERIMENTAL SETTINGS

**AIGI Detectors.** We evaluate our confidence model on six representative AIGI detectors: CD (Wang et al., 2020), ND (Gragnaniello et al., 2021), NPR (Tan et al., 2024b), FreqNet (Tan et al., 2024a), SAFE (Li et al., 2024b) and C2P (Tan et al., 2025). All models except C2P were retrained on the ForenSynths dataset using the four-class training protocol (*i.e.*, cars, cats, chairs, and horses). Since NPR, FreqNet, and SAFE were originally trained on pristine images without robustness enhancements, we incorporated two distortions as augmentations during their training, *i.e.*, JPEG

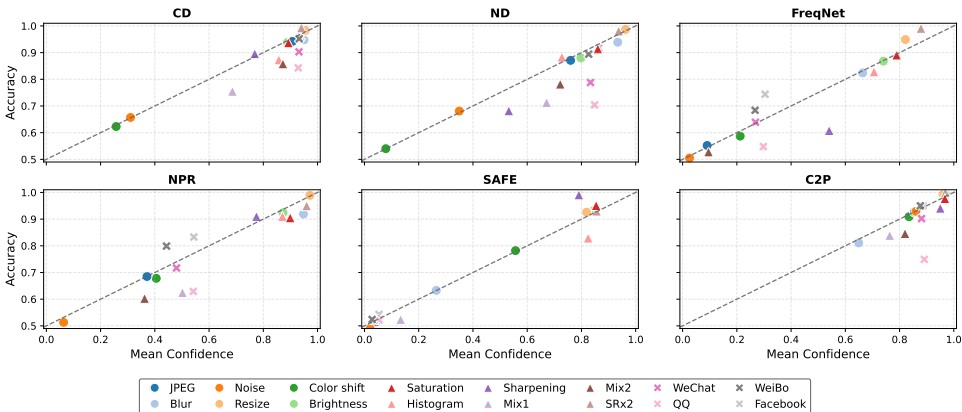

Figure 4: Correlation between average confidence and detection accuracy. ○: seen distortions; △/×: unseen single/platform-compression distortions.

compression and blurring. The JPEG compression quality was sampled from $[30, 100]$, and the blurring kernel size varied in $[0, 3]$. Both augmentations were applied with a probability of 10%. The resulting models were denoted $NPR^+$, $FreqNet^+$ and $SAFE^+$. Unless stated otherwise, experiments were conducted with the trained models CD, ND, $NPR^+$, $FreqNet^+$, $SAFE^+$. C2P provides no training code; we therefore used its released pretrained weights, denoted as $C2P^*$. Meanwhile, to validate the generality of our approach, we transfer the data domain from GANs to diffusion models. Specifically, we retrain our detector on a diffusion-model dataset and evaluate the proposed method under the same experimental settings. Detailed results and analyses are provided in the Appendix I.

**Datasets.** Eight distortion families were used when training DACOM, including JPEG, Blur, Noise, Resize, Color warming, Color cooling, Brighten, and Darken. Each distortion has an associated test split, forming the **Seen Distortion test sets**. To evaluate the generalization ability, we constructed **Unseen Distortion test sets** comprising ten distortion types that never appear in training. Additionally, to evaluate the screening performance, we also included an **Evaluation-dataset** and a **Cross-dataset**. More details about datasets are provided in Appendix D.

**Implementation of DACOM.** To form the training-time supervision, we experimented with four FR-IQA metrics, including SSIM (Hore & Ziou, 2010), MS-SSIM (Sara et al., 2019), FSIM (Zhang et al., 2011), and DISTS (Ding et al., 2020), producing four variants of our model. The models are referred to as $DACOM_{SSIM}$, $DACOM_{MS-SSIM}$, $DACOM_{FSIM}$ and $DACOM_{DISTS}$. In the proposed confidence model, we employ **QualiCLIP**, a NR-IQA model that operates entirely in a self-supervised manner without relying on Mean Opinion Score (MOS) for supervision, as our **Image Quality Encoder**. We adopt the feature extractor from **ARNIQA** (Agnolucci et al., 2024b) as our **Distortion Type Encoder**, considering its strong capability in identifying distortion types. Further details regarding the selection of Distortion Type Encoder are discussed in the Appendix E. More details of the experimental configuration of DACOM are provided in the Appendix F.

**Comparison Methods.** We compare DACOM against three categories of methods in our experiments: (1) FR-IQA methods: SSIM, MS-SSIM, FSIM, and DISTS; for clarity, we prefix full-reference metrics with "FR-". (2) NR-IQA methods: TOPIQ (Chen et al., 2024b), ARNIQA, and QualiCLIP. (3) Post-hoc logit calibration (Guo et al., 2017). For the logit calibration baseline, the same training set as used for the confidence model is employed to normalize the output of each classifier, and the detector's confidence score is given by $|logit - 0.5|$.

## 5.2 Correlation between Estimated Confidence and Detector Accuracy

We first verify whether DACOM can faithfully predict the confidence of a given detector. Using the conventional linear- and rank-correlation measures, DACOM achieves a Pearson linear correlation coefficient (PLCC) of **97.73%** and a Spearman rank correlation coefficient (SRCC) of **94.01%** on the test set, confirming the effectiveness of our strategy. Complete results are reported in Appendix Table 8. Figure 4 plots average confidence against detection accuracy for both the seen and unseen

Table 1: Performance (%) of multi-detector fusion on the **Seen Distortion test sets**. For every image we conduct Top-1 routing, using the prediction from the detector whose associated DACOM yields the highest confidence score. "**Average**" is the mean value over all eight distortion subsets, whereas "**Worst**" corresponds to the poorest performance observed on any single subset.

| Method | JPEG | | Blur | | Noise | | Resize | | Color shift | | Brightness | | **Average** | | **Worst** | |
|---|---|---|---|---|---|---|---|---|---|---|---|---|---|---|---|---|
| | Acc | AP | Acc | AP | Acc | AP | Acc | AP | Acc | AP | Acc | AP | Acc | AP | Acc | AP |
| CD | 94.20 | 99.28 | 94.80 | 99.36 | 65.70 | 93.37 | 98.30 | 99.90 | 62.30 | 90.75 | 93.80 | 99.74 | 84.85 | 97.07 | 62.30 | 90.75 |
| ND | 87.10 | 95.88 | 93.90 | 99.01 | 68.10 | 80.71 | 98.70 | 99.93 | 54.00 | 66.79 | 88.00 | 98.33 | 81.63 | 90.11 | 54.00 | 66.79 |
| FreqNet[+] | 55.20 | 67.40 | 82.40 | 90.66 | 50.50 | 50.87 | 94.90 | 98.15 | 58.70 | 64.48 | 86.80 | 95.02 | 71.42 | 77.76 | 50.50 | 50.87 |
| NPR[+] | 68.50 | 79.99 | 91.90 | 97.85 | 51.30 | 51.34 | 98.90 | 99.98 | 67.80 | 76.86 | 92.40 | 98.85 | 78.47 | 84.15 | 51.30 | 51.34 |
| SAFE[+] | 52.30 | 55.11 | 63.30 | 79.31 | 49.00 | 48.06 | 92.60 | 99.21 | 78.20 | 85.73 | 93.20 | 98.39 | 71.43 | 77.63 | 49.00 | 48.06 |
| C2P[*] | 92.70 | 97.98 | 81.10 | 94.15 | **92.90** | **99.27** | 99.20 | **100.00** | 90.90 | 99.05 | 99.00 | **99.99** | 92.63 | 98.41 | 81.10 | 94.15 |
| Logit Calib. | 92.20 | 98.07 | 92.70 | 97.59 | 64.40 | 55.46 | **99.90** | **100.00** | 89.20 | 92.09 | 99.10 | 99.78 | 89.58 | 90.49 | 64.40 | 55.46 |
| DACOM_SSIM | 94.00 | **99.28** | 94.70 | 98.91 | **92.90** | **99.27** | 99.40 | 99.90 | **90.90** | **99.05** | 98.90 | 99.98 | 95.13 | 99.42 | **90.90** | **99.05** |
| DACOM_MS-SSIM | 94.20 | 99.19 | 95.30 | 98.92 | **92.90** | **99.27** | 99.40 | 99.99 | **90.90** | **99.05** | 98.90 | 99.98 | 95.27 | 99.40 | **90.90** | **99.05** |
| DACOM_FSIM | 94.20 | 99.13 | **95.60** | **99.38** | **92.90** | **99.27** | 99.60 | 99.99 | **90.90** | **99.05** | 99.20 | **99.99** | 95.40 | **99.47** | **90.90** | **99.05** |
| DACOM_DISTS | **94.30** | 99.13 | 95.50 | 99.38 | **92.90** | **99.27** | 99.60 | 99.99 | **90.90** | **99.05** | 99.00 | **99.99** | 95.37 | **99.47** | **90.90** | **99.05** |

Table 2: Performance (%) of multi-detector fusion on the **Unseen Distortion test sets**.

| Method | Histogram | | Saturation | | Sharpening | | Mix1 | | Mix2 | | WeChat | | QQ | | Weibo | | Facebook | | SRx2 | | **Average** | | **Worst** | |
|---|---|---|---|---|---|---|---|---|---|---|---|---|---|---|---|---|---|---|---|---|---|---|---|---|
| | Acc | Ap | Acc | Ap | Acc | Ap | Acc | Ap | Acc | Ap | Acc | Ap | Acc | Ap | Acc | Ap | Acc | Ap | Acc | Ap | Acc | Ap | Acc | Ap |
| CD | 87.30 | 97.72 | 93.70 | 99.75 | 89.60 | 99.32 | 75.50 | 88.92 | 85.80 | 94.66 | 90.30 | 99.12 | **84.30** | **98.86** | 95.20 | 99.61 | 95.90 | **99.74** | 99.30 | 99.99 | 89.69 | 97.77 | 75.50 | 88.92 |
| ND | 88.30 | 98.83 | 91.40 | 98.90 | 68.20 | 89.58 | 71.30 | 79.61 | 78.20 | 85.69 | 78.70 | 92.93 | 70.30 | 90.24 | 89.40 | 97.17 | 91.70 | 98.32 | 98.10 | 99.70 | 82.56 | 93.02 | 68.20 | 89.58 |
| FreqNet[+] | 82.80 | 93.29 | 89.10 | 94.83 | 60.80 | 75.10 | 55.00 | 53.34 | 52.80 | 59.97 | 63.90 | 67.83 | 54.80 | 58.34 | 68.30 | 74.04 | 74.30 | 81.25 | 99.10 | 99.91 | 70.09 | 75.79 | 52.80 | 53.34 |
| NPR[+] | 91.00 | 98.57 | 90.50 | 96.24 | 91.00 | 97.22 | 62.50 | 69.07 | 60.30 | 67.41 | 71.70 | 81.10 | 62.90 | 72.39 | 79.90 | 88.31 | 83.30 | 91.35 | 95.00 | 99.46 | 78.81 | 86.11 | 60.30 | 67.41 |
| SAFE[+] | 82.90 | 95.26 | 95.10 | 99.23 | **99.10** | **99.94** | 52.40 | 50.48 | 51.60 | 50.85 | 52.20 | 58.32 | 52.20 | 54.77 | 52.40 | 58.94 | 53.40 | 60.43 | 93.00 | 98.56 | 68.52 | 72.68 | 51.60 | 50.48 |
| C2P[*] | 99.80 | 100.00 | **97.70** | 99.97 | 94.10 | **99.99** | 83.90 | 92.08 | 84.60 | 93.78 | 90.20 | 95.78 | 74.90 | 88.94 | 95.00 | 98.61 | 95.00 | 98.30 | **100.00** | **100.00** | 91.52 | 96.75 | 74.90 | 88.94 |
| Logit Calib. | **99.90** | **100.00** | 97.50 | 99.83 | 98.70 | 99.24 | 77.20 | 68.66 | 82.50 | 92.07 | 85.50 | 85.03 | 78.40 | 83.56 | 88.30 | 86.58 | 95.10 | 96.06 | **100.00** | **100.00** | 90.31 | 90.95 | 77.20 | 68.66 |
| DACOM_SSIM | 99.80 | **100.00** | 97.00 | 99.35 | 94.10 | **99.99** | 85.20 | 93.24 | 86.90 | 95.23 | 90.70 | **98.93** | 84.00 | 98.26 | 95.30 | 99.55 | 96.10 | 99.65 | 97.00 | 99.33 | 92.70 | 98.53 | **84.00** | 93.24 |
| DACOM_MS-SSIM | 99.80 | **100.00** | 97.60 | 99.77 | 94.10 | **99.99** | 83.20 | 91.07 | 86.70 | 95.03 | 90.80 | 98.79 | 83.40 | 97.28 | **95.60** | 99.54 | 96.10 | 99.49 | 98.50 | 99.94 | 92.58 | 98.04 | 83.20 | 91.07 |
| DACOM_FSIM | 99.80 | **100.00** | **97.70** | 99.97 | 94.10 | **99.99** | **85.40** | **93.33** | 86.30 | 95.11 | 90.80 | **98.93** | 83.80 | 97.75 | 95.50 | **99.62** | **96.20** | 99.55 | 99.70 | 99.80 | **92.93** | **98.40** | 83.80 | **93.33** |
| DACOM_DISTS | 99.80 | **100.00** | **97.70** | 99.87 | 94.10 | **99.99** | 84.90 | 93.16 | **87.00** | **95.21** | 90.70 | 98.79 | 83.20 | 97.45 | 95.40 | 99.57 | 95.90 | 99.47 | 99.70 | 99.80 | 92.84 | 98.33 | 83.20 | 93.16 |

distortion test sets. A strong positive correlation is evident for all the involved detetors. Detailed quantitative results are provided in the Appendix G.

## 5.3 RESULTS OF TOP-1 ROUTING IN MULTI-DETECTOR SCENARIO

We next test whether the estimated confidence can guide an on-the-fly choice among multiple detectors. The six detectors described in Section 5.1 are each equipped with its own DACOM. Given an image, six confidence scores are computed and the prediction of the detector with the highest score is returned, denoted as "Top-1 routing" fusion. This strategy is compared against two baselines: (1) using a single detector's output; (2) using the calibrated logit-based confidence to perform Top-1 routing. Results on the Seen Distortion test sets (Table 1) show that Top-1 routing with DACOM consistently outperforms the single best detector, demonstrating the effectiveness of our confidence model in identifying the most suitable detector for each input. Notably, our best result surpasses logit-based Top-1 routing by 5.82% in mean accuracy (Acc) and 8.98% in mean average precision (AP). On unseen distortion types (Table 2), our approach still outperforms logit-based fusion by 2.62% in mean accuracy and 8.45% in mean AP, showcasing its robustness and generalization capabilities, especially under complex distortion conditions such as social media transmission.

## 5.4 RESULTS OF CONFIDENCE-BASED FILTERING

We finally examine whether DACOM can serve as a reliable basis for rejecting uncertain predictions. Experiments are carried out on the Evaluation-dataset and the Cross-dataset, and each image undergoes a single, randomly selected distortion. For every detector we rank all test images by the confidence produced by its corresponding DACOM. Starting from the full set, we iteratively discard the lowest-confidence $p\%$ of samples and compute the Balanced Accuracy (BAcc) and Equal Error Rate (EER) on the retained subset. Tables 3 (Evaluation-dataset) and 4 (Cross-dataset) show that, at every filtering rate, DACOM achieves higher BAcc and lower EER compared to the logit-based confidence baseline. The results for random multiple distortions are provided in Appendix H and exhibit similar trend. Furthermore, we plot the Risk-Coverage (RC) curve (Figure 5). As more low-confidence samples are filtered out, the risk on the retained set monotonically decreases, demonstrating the effectiveness of confidence-based filtering.

Table 4: Results on Cross-dataset Single-Distortion Filtering (%).

| Method | Distortion | | Filtering Proportion | | | | | | | | | | | |
|---|---|---|---|---|---|---|---|---|---|---|---|---|---|---|
| | BAcc ↑ | EER ↓ | 0.05 | | 0.10 | | 0.15 | | 0.20 | | 0.30 | | 0.40 | |
| Topiq | | | 70.65 | 26.70 | 71.05 | 26.11 | 71.42 | 25.54 | 71.81 | 24.94 | 72.53 | 23.85 | 73.20 | 22.86 |
| ARNIQA | 70.19 | 27.26 | 70.24 | 27.22 | 70.33 | 27.10 | 70.38 | 26.91 | 70.42 | 26.86 | 69.98 | 27.06 | 69.83 | 26.97 |
| QualiCLIP | | | 70.91 | 26.33 | 71.50 | 25.57 | 71.96 | 25.02 | 72.38 | 24.49 | 72.92 | 23.85 | 73.25 | 23.49 |
| Logit Calib. | | | 70.81 | 27.16 | 71.38 | 27.08 | 71.92 | 27.14 | 72.29 | 26.98 | 73.06 | 27.07 | 73.71 | 27.50 |
| FR-SSIM | | | 70.82 | 26.54 | 71.41 | 25.80 | 71.97 | 25.16 | 72.51 | 24.65 | 73.31 | 23.79 | 74.16 | 22.89 |
| FR-MS-SSIM | 70.19 | 27.26 | 70.66 | 26.70 | 71.24 | 26.01 | 71.82 | 25.34 | 72.46 | 24.61 | 73.50 | 23.55 | 74.49 | 22.46 |
| FR-FSIM | | | 70.67 | 26.66 | 71.20 | 26.00 | 71.80 | 25.27 | 72.35 | 24.60 | 73.35 | 23.54 | 74.44 | 22.41 |
| FR-DISTS | | | 70.94 | 26.32 | 71.75 | 25.30 | 72.48 | 24.47 | 73.14 | 23.70 | 74.51 | 22.18 | 75.73 | 20.96 |
| DACOM$_{SSIM}$ | | | 71.01 | 26.03 | 71.86 | 24.92 | 72.87 | 23.89 | 73.70 | _23.12_ | 75.37 | 21.71 | 77.06 | 20.09 |
| DACOM$_{MS-SSIM}$ | 70.19 | 27.26 | _71.05_ | **25.94** | _71.95_ | 24.85 | _72.94_ | 23.85 | _73.74_ | 23.14 | 75.39 | 21.79 | 77.09 | 19.99 |
| DACOM$_{FSIM}$ | | | **71.06** | 26.02 | **71.97** | 24.87 | **72.97** | **23.88** | **73.80** | 23.09 | _75.46_ | _21.73_ | _77.22_ | _19.89_ |
| DACOM$_{DISTS}$ | | | **71.06** | _25.97_ | _71.95_ | **24.81** | 72.83 | 23.88 | 73.71 | 23.09 | 75.54 | **21.69** | **77.32** | **19.87** |

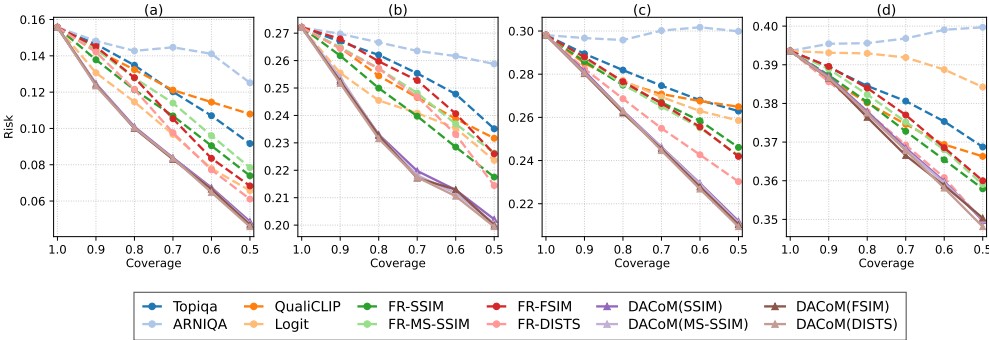

Figure 5: Risk-Coverage (RC) curves with risk defined as $1 - \text{BAcc}$ (x-axis: coverage, y-axis: risk). (a) and (b) show results on the evaluation dataset with single- and multi-distortion, while (c) and (d) present Cross-dataset results under single- and multi-distortion.

## 5.5 ABLATION STUDY

To disentangle the contribution of each design choice inside DACOM, we evaluate the DACOM encoder design and the choice of the Image Quality Encoder $\phi_{IQ}$. As summarized in Appendix J, incorporating distortion-sensitive features from $\phi_{IQ}$ consistently improves regression. **The best configuration is using QualiCLIP as Image Quality Encoder and including a Distortion Type Encoder**, achieving the strongest overall performance with the average **PLCC** of **97.66%** and **SRCC** of **93.97%** across detectors. We conducted a comparative study on the mapping functions for transforming detectability labels, and the detailed results are reported in Appendix Table 18. Among the considered alternatives, our mapping function achieves the best performance.

Table 3: Evaluation-dataset Single-Distortion Filtering (%). Averages over multiple detectors. "Distortion" reports BAcc and EER on distorted sets; "Filtering Proportion" is the fraction of samples removed. Best and second-best are **bold** and underlined, respectively.

| Method | Distortion | | Filtering Proportion | | | | | | | | | | | |
|---|---|---|---|---|---|---|---|---|---|---|---|---|---|---|
| | BAcc ↑ | EER ↓ | 0.05 | | 0.10 | | 0.15 | | 0.20 | | 0.30 | | 0.40 | |
| Topiq | | | 84.89 | 14.51 | 85.35 | 14.19 | 85.90 | 13.84 | 86.52 | 13.21 | 87.98 | 11.63 | 89.30 | 10.13 |
| ARNIQA | 84.43 | 14.79 | 84.86 | 14.50 | 85.18 | 14.40 | 85.46 | 14.38 | 85.73 | 14.27 | 85.53 | 14.86 | 85.89 | 14.45 |
| QualiCLIP | | | 85.02 | 14.27 | 85.69 | 13.67 | 86.23 | 13.15 | 86.76 | 12.64 | 87.89 | 11.58 | 88.55 | 10.86 |
| Logit Calib. | | | 85.86 | 14.19 | 86.93 | 13.53 | 87.85 | 13.30 | 88.54 | 12.73 | 90.33 | 11.61 | 92.21 | 10.73 |
| FR-SSIM | | | 85.34 | 13.97 | 86.22 | 13.21 | 87.10 | 12.36 | 87.87 | 11.44 | 89.31 | 10.32 | 90.95 | 8.80 |
| FR-MS-SSIM | 84.43 | 14.79 | 84.90 | 14.46 | 85.63 | 13.81 | 86.49 | 13.04 | 87.22 | 12.30 | 88.60 | 10.89 | 90.40 | 9.22 |
| FR-FSIM | | | 84.93 | 14.38 | 85.49 | 14.08 | 86.30 | 13.34 | 87.20 | 12.38 | 89.46 | 10.05 | 91.65 | 8.05 |
| FR-DISTS | | | 84.98 | 14.39 | 85.79 | 13.76 | 86.67 | 12.88 | 87.85 | 11.64 | 90.23 | 9.29 | 92.28 | 7.46 |
| DACOM$_{SSIM}$ | | | 86.01 | 13.13 | 87.53 | 11.54 | 88.99 | 10.51 | 89.91 | 9.24 | 91.61 | _7.11_ | 93.26 | 5.35 |
| DACOM$_{MS-SSIM}$ | 84.43 | 14.79 | _86.04_ | 13.10 | 87.56 | 11.57 | _89.03_ | **10.41** | _89.98_ | _9.18_ | _91.67_ | **7.06** | 93.32 | _5.26_ |
| DACOM$_{FSIM}$ | | | _86.04_ | _13.05_ | _87.60_ | _11.43_ | **89.05** | 10.44 | **90.00** | **9.14** | **91.70** | 7.11 | _93.35_ | 5.27 |
| DACOM$_{DISTS}$ | | | **86.17** | **12.96** | **87.66** | **11.41** | **89.05** | _10.43_ | 89.97 | 9.21 | 91.65 | 7.15 | **93.52** | **5.12** |

## 6 CONCLUSION

This work takes the problem of AI-generated image (AIGI) detection as a test-bed and offers the first systematic analysis of detector reliability in the presence of common, real-world distortions. We show that the raw outputs of detectors cannot convey sample-level confidence once images are degraded, and we introduce DACOM, a distortion-aware confidence model, to enable detectors to output a confidence score for each image under distortion. Experiments across diverse distortions and detectors support three findings: (i) FR-IQA-based distortion levels align more strongly with forensic performance than no-reference IQA (NR-IQA) scores; (ii) distortion effects are detector- and type-dependent—even at matched intensities—revealing systematic interaction patterns; and (iii) the proposed confidence model enables effective sample-level filtering and multi-detector fusion, improving overall reliability. Despite these advances, DACOM exhibits degraded performance in cross-dataset evaluation, reflecting its sensitivity to changes in the data source distribution. To address this, we will explore more general and shift-resilient formulations of distortion-aware, detector-conditioned reliability modeling, strengthening the framework as a foundation for reliability estimation under real-world degradations.

## ACKNOWLEDGMENTS

This work was supported in part by NSFC (Grant U23B2022, U22B2047, 62572325, 62371301), Shenzhen R&D Program (Grant JCYJ20250604181211016, SYSPG20241211174032004), Guangdong Basic and Applied Basic Research Foundation (Grant 2025A1515010234).

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

# A  IQA METRICS AND DETECTION PERFORMANCE

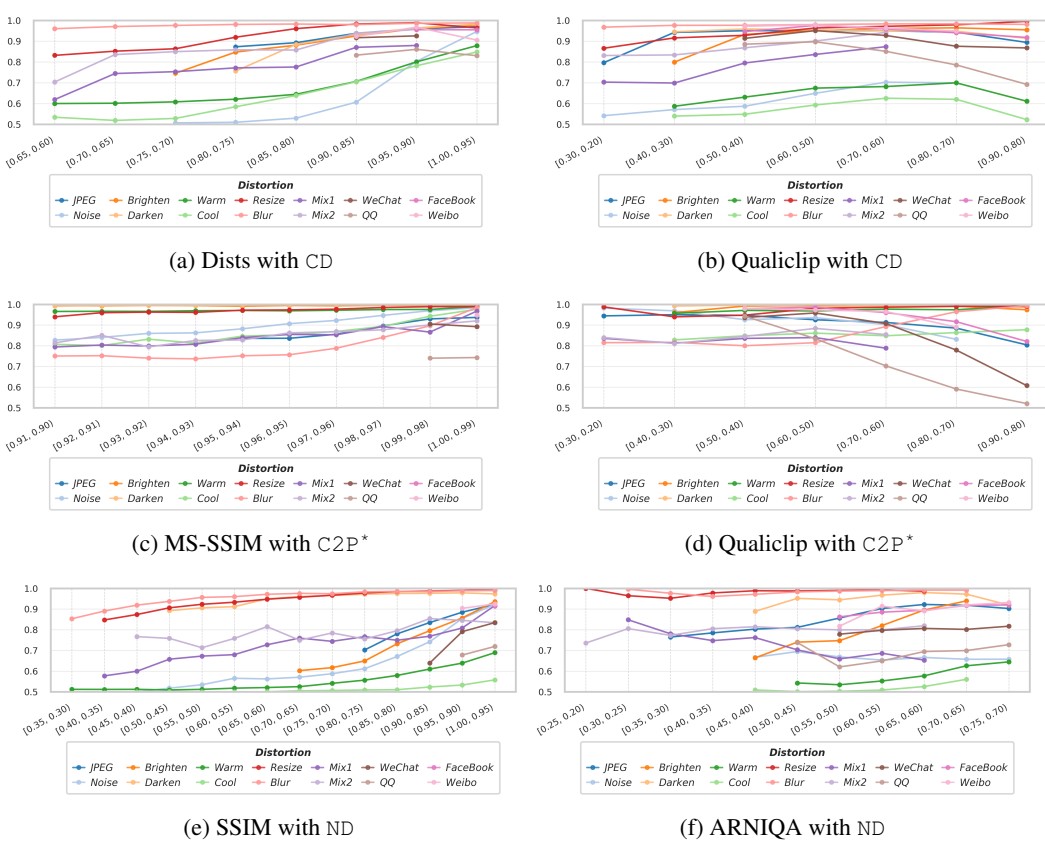

(a) Dists with CD

(b) Qualiclip with CD

(c) MS-SSIM with C2P*

(d) Qualiclip with C2P*

(e) SSIM with ND

(f) ARNIQA with ND

Figure 6: Relationship between IQA Metrics and Model Detection Performance.

We present the correlation between IQA scores and model detection performance for a broader set of distortion types in Figure 6. The results indicate a consistently strong correlation across diverse distortion categories.

# B  DISTORTION TYPES, FR-IQA, AND MODEL LOGIT DISTRIBUTIONS

We visualize the drift in output logit distributions of various detectors as the distortion severity increases (indicated by a decrease in FR-IQA metrics). As shown in the Figure 7, we observe that different detectors exhibit distinct distributional shifts under the same distortion type and severity.

In Figure 7 (a)-(c), under Gaussian white noise with increasing intensity, both CD and SAFE⁺ exhibit a shift of the fake distribution toward the real distribution, whereas NPR⁺ shows the opposite trend. In Figure 7 (d)-(f), under color shift distortions, CD shows a shift of the fake distribution toward the real one; FreqNet⁺ displays a convergence of both distributions toward the decision boundary; and C2P behaves oppositely to CD. In Figure 7 (g)-(i), under distortions introduced by four different social media platforms, CD consistently shows the fake distribution shifting toward the real one; NPR⁺ exhibits a convergence of both distributions toward the center; and SAFE⁺ demonstrates a complete shift of the real distribution into the domain of fakes. These results indicate that under identical distortion conditions, different detectors are affected dissimilarly, leading to varied patterns of misclassification.

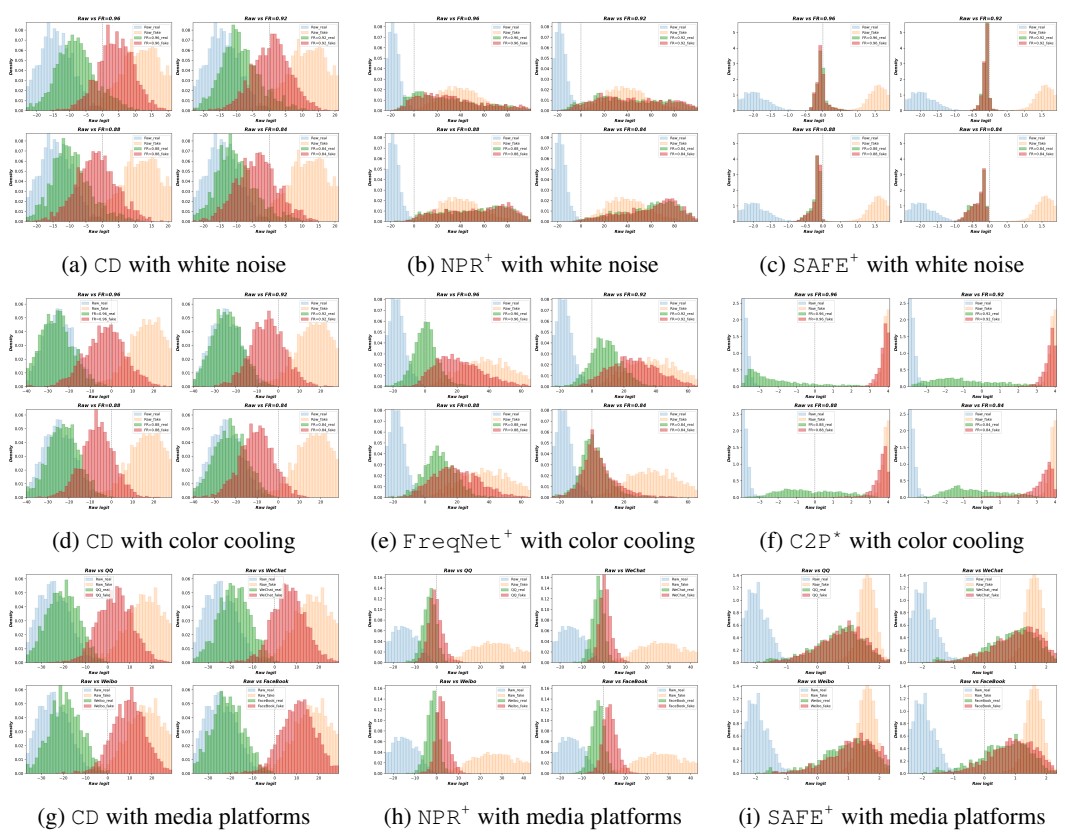

(a) CD with white noise  (b) NPR$^+$ with white noise  (c) SAFE$^+$ with white noise

(d) CD with color cooling  (e) FreqNet$^+$ with color cooling  (f) C2P$^*$ with color cooling

(g) CD with media platforms  (h) NPR$^+$ with media platforms  (i) SAFE$^+$ with media platforms

Figure 7: The logit distribution shifts under the influence of distortion. Blue/yellow: clean real/fake; green/red: distorted real/fake.

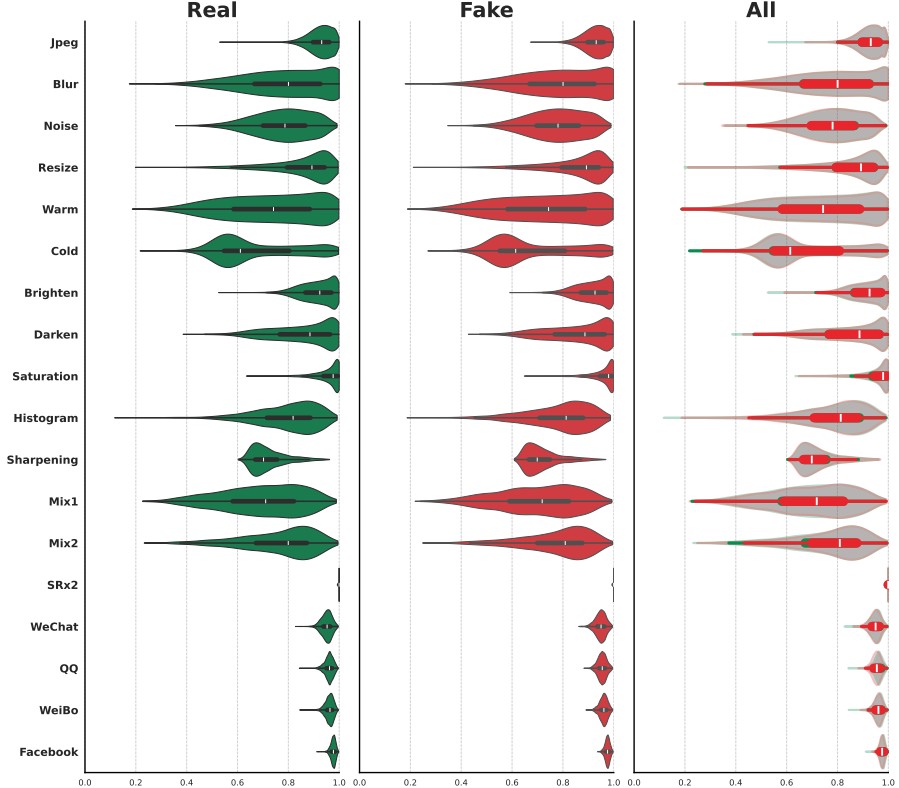

Figure 8: Distribution of FR-IQA Scores Across Distortion Types (SSIM).

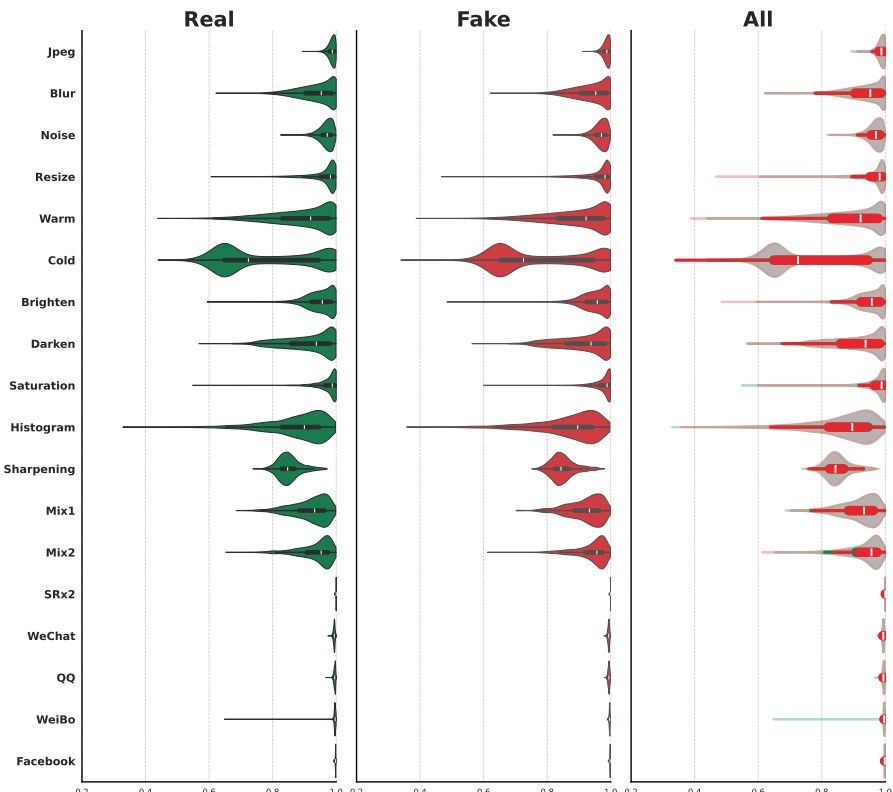

Figure 9: Distribution of FR-IQA Scores Across Distortion Types (MS-SSIM).

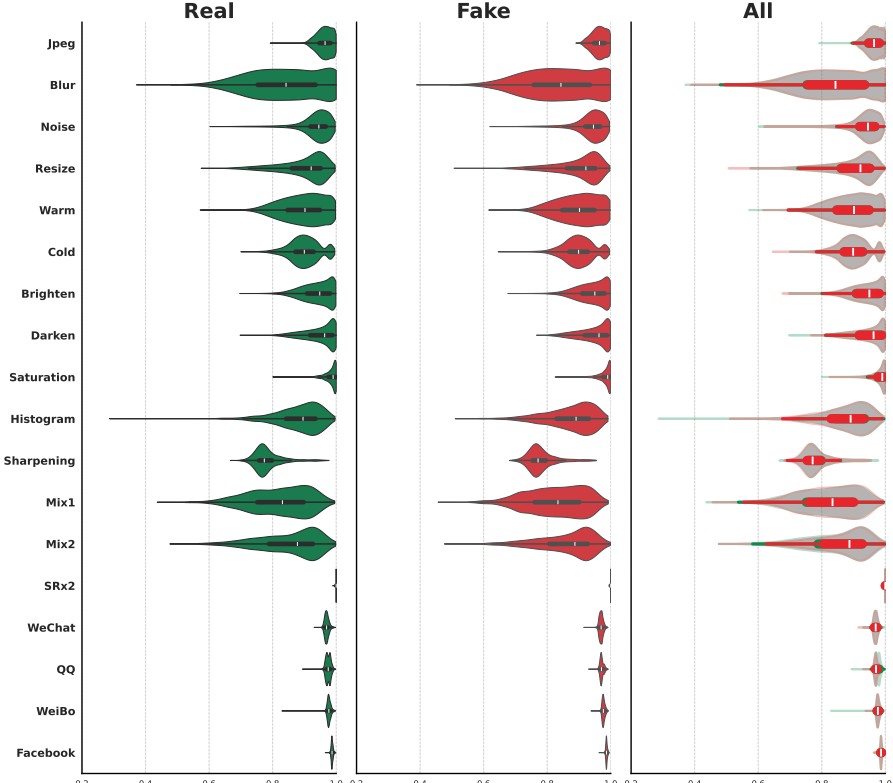

Figure 10: Distribution of FR-IQA Scores Across Distortion Types (FSIM).

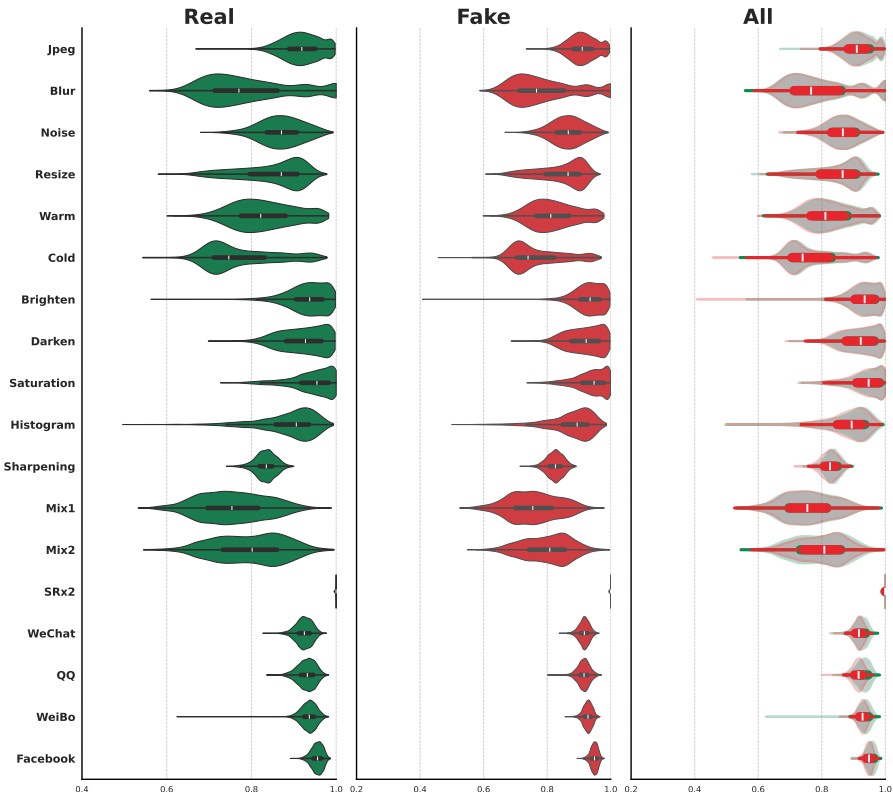

Figure 11: Distribution of FR-IQA Scores Across Distortion Types (DISTS).

## C  DISTRIBUTION OF IMAGE QUALITY ASSESSMENT SCORES UNDER DISTORTION

We visualize the distribution ranges of full-reference image quality assessment (FR-IQA) scores across different datasets of distortion types, as shown in Figures 8–11. The distortion types from top to bottom are: JPEG, Blur, Noise, Resize, Color warming, Color cooling, Brighten, Darken, Saturation, Histogram, Sharpening, Mix1, Mix2, SRx2, WeChat, QQ, WeiBo, and Facebook. For each distortion type, we present the score distributions for real images, fake images, and their combination. Several key observations can be made. First, the score ranges vary considerably across distortion types. For instance, the scores for JPEG compression consistently fall within [0.8, 1.0] across all four FR-IQA methods, and a similar concentration is observed for the four social media platforms (WeChat, QQ, WeiBo, Facebook). Second, the distributions for real and fake images are highly consistent for each distortion type. This indicates that the FR-IQA methods perceive the overall quality of real and fake images similarly, without being significantly affected by potential artifacts. This property ensures that our approach treats both real and fake data fairly.

## D  DATASET DETAILS

We randomly sample 4,000 images (with an equal number of real and fake samples) from the Pro-GAN test set (totaling 8,000 images) (Wang et al., 2020) to form the base dataset for the confidence model. Based on this, we design eight common distortion types: JPEG compression, blur, additive white Gaussian noise, resize, darkening, brightening, and two types of color shift. Each distortion type is applied at 10 intensity levels, generating 40,000 distorted images per type (covering all intensities of that distortion), resulting in a total of 320,000 distorted images for training and evaluating the regression performance of the confidence model. During training and evaluation, the original 4,000 images are split into 2,500/500/1,000 for the training, validation, and test sets, respectively. After applying distortions, the overall dataset sizes become 202,500, 40,500, and 81,000 for these three subsets. The remaining 4,000 images from the ProGAN test set are reserved as an independent **Evaluation-dataset** and are not involved in the training or validation of the confidence model.

Table 5: Dataset construction with distortions, number of samples, and sources.

| Dataset | Distortion | Number | Source |
|---|---|---|---|
| Training | JPEG, Blur, Noise, Resize, | 202,500 | 2500 samples from ForenSynths ProGan |
| Validation | Color warming, Color cooling, | 40,500 | 500 samples from ForenSynths ProGan |
| Test | Brighten, Darken | 81,000 | 1000 samples from ForenSynths ProGan |
| Seen Distortion test sets | JPEG | 1,000 | |
| | Blur | 1,000 | |
| | Noise | 1,000 | |
| | Resize | 1,000 | |
| | Color shift | 1,000 | |
| | Brightness | 1,000 | |
| Unseen Distortion test sets | Mix1 | 1,000 | From the same data split as the test |
| | Mix2 | 1,000 | |
| | Saturation | 1,000 | |
| | Histogram | 1,000 | |
| | Sharpening | 1,000 | |
| | SR ($2\times$ upscaling) | 1,000 | |
| | QQ | 1,000 | |
| | WeChat | 1,000 | |
| | Weibo | 1,000 | |
| | Facebook | 1,000 | |
| Evaluation-dataset single distortion | randomly select one distortion type | 4,000 | 4000 samples from ForenSynths Progan |
| Evaluation-dataset multiple distortions | randomly combine 2–4 distortion types | 4,000 | |
| Cross-dataset single distortion | randomly select one distortion type | – | Subsets for which the detector achieves reasonable performance (accuracy > 0.75) **ForenSynths:** {Stylegan, Stylegan2, Biggan, Cyclegan, Stargan, Gaugan, Deepfake} **Universal:** {glide_100_10, glide_100_27, glide_50_27, DALLE, ldm_100, ldm_200, ldm_200_cfg}, **GenImage:** {ADM, BigGAN, glide, Midjourney, SD_v14, SD_v15, VQDM}, |
| Cross-dataset multiple distortions | randomly combine 2–4 distortion types | – | **DiTFake:** {FLUX, PixArt, SD3} |

To further evaluate the model's generalization capability, we construct test sets containing both seen and Unseen Distortions. **Seen Distortion test sets.** We merge **Brighten** and **Darken** into **Brightness**, and **Color warming** and **Color cooling** into **Color shift**. For each of the six distortion types mentioned above, we apply a randomly selected intensity to every image in the test set (original images), resulting in six corresponding distorted subsets. **Unseen Distortion test sets.** This includes the following ten categories: (1) **Mix1**: Randomly selects 2–4 distortion types from {JPEG, Blur, Noise, Resize} and combines them randomly; (2) **Mix2**: Similarly selects 2–4 types, but fixes JPEG as the final distortion step (with others applied randomly before).; (3) **QQ** platform transmission; (4) **WeChat** platform transmission; (5) **Weibo** platform transmission; (6) **Facebook** platform transmission; (7) **Saturation adjustment**; (8) **Histogram equalization**; (9) **Image sharpening**: using the unsharp mask (USM) enhancement algorithm; (10) **Super-Resolution ($2\times$ upscaling)**: performed with a Transformer-based architecture (Zhang et al., 2022).

Table 6: The types of distortions and their corresponding intensities used in the training set. "Parameter Range" indicates the closed interval "[min, max]" for a single distortion.

| Distortion Type | Parameter Range | Step |
|---|---|---|
| JPEG | $[24, 96]$ | 8 |
| Gaussian blur | $[0.3, 3.0]$ | 0.3 |
| Noise | $[0.0002, 0.002]$ | 0.0002 |
| Resize | $[0.20, 0.92]$ | 0.08 |
| Color warming | $[1.05, 1.5]$ | 0.05 |
| Color cooling | $[0.5, 0.95]$ | $-0.05$ |
| Brighten | $[0.05, 0.5]$ | 0.05 |
| Darken | $[-0.5, -0.05]$ | $-0.05$ |

Additionally, in the sample filtering experiments, we introduce Cross-dataset evaluation based on the Evaluation-dataset. Specifically, from the three dataset collections–ForenSynths (Wang et al., 2020), Universal (Ojha et al., 2023), GenImage (Zhu et al., 2023) and DiTFake (Li et al., 2024b)–

Table 7: The types of distortions and their corresponding intensities applied in the test set. We only list distortion types with tunable parameters; Histogram, Super-Resolution (2×), QQ, WeChat, Weibo, and Facebook have no adjustable parameters and are omitted. For "Mix1" and "Mix2", parameter ranges are not fixed; instead, each component distortion randomly samples its intensity from its own range. The "Sharpening" distortion uses the Unsharp Mask algorithm internally, controlled by radius (blur extent) and amount (enhancement strength).

| Distortion Type | Parameter Range | Step |
|---|---|---|
| JPEG | $[24, 100]$ | 2 |
| Gaussian blur | $[0.15, 5.0]$ | 0.15 |
| Noise | $[0.0001, 0.002]$ | 0.0001 |
| Resize | $[0.5, 1.5]$ | 0.2 |
| Color shift | $[0.5, 1.5]$ | 0.2 |
| Saturation | $[0.1, 1.9]$ | 0.2 |
| Brightness | $[-0.5, 0.5]$ | 0.2 |
| Sharpening | radius: $[1, 5]$, amount: $[0.5, 3]$ | Random within ranges |
| Mix1 | Random 2–4 from {JPEG, Blur, Noise, Resize} | Per-distortion random |
| Mix2 | Random 1–3 from above set + JPEG | Per-distortion random |

we select subsets on which the detector achieves reasonable performance (accuracy $> 0.75$). Note that the number of selected subsets may vary across detectors due to performance differences. We then apply single distortions (randomly selecting one distortion type and intensity) and multiple distortions (randomly combining 2–4 distortion types) to these subsets, limiting the distortions to the eight types already encountered by the confidence model. Table 5 presents the full details of our datasets, while Tables 6 and 7 summarize the distortion types and their corresponding parameter ranges used in the training and test sets, respectively.

# E    DISTORTION TYPE CLASSIFICATION MODEL

To investigate whether features from NR-IQA methods can be used for explicit distortion identification, we report the distortion-type classification results of several NR-IQA methods (**ARNIQA**, **QualiCLIP**, **Topiq** (Chen et al., 2024b), and **BRISQUE** (Mittal et al., 2012)). We freeze the feature extractors of each NR-IQA model and directly feed their output features into a classification head for distortion-type classification.

We sample 9,600 distorted images from the distorted dataset, covering eight types of distortions: JPEG compression, noise, resizing, blur, darken, brighten, and two types of color shifts ("Sat Down" corresponding to Cool and "Sat Up" corresponding to Warm). Among these, 8,000 images are used to train the classifier, and 1,600 are reserved for testing. We visualize the confusion matrices corresponding to the best classification performance achieved by each NR-IQA method over 100 epochs, as shown in the figure 12. "Sat Down" and "Sat Up" refer to the two specific types of color shifts. It can be intuitively observed that **ARNIQA** achieves the best classification accuracy, followed by **QualiCLIP**. Therefore, we select **ARNIQA** as the Distortion-Type Encoder in our method section.

Table 8: Regression performance of **DACOM** across detectors trained with different FR-IQA (SSIM, MS-SSIM, FSIM, DISTS). Evaluated by PLCC and SRCC on the test set; all methods yield consistently high scores, supporting the proposed FR-guided supervision strategy.

| Method | CD | | ND | | NPR[+] | | FreqNet[+] | | SAFE[+] | | C2P[*] | | Average | |
|---|---|---|---|---|---|---|---|---|---|---|---|---|---|---|
| | PLCC | SRCC | PLCC | SRCC | PLCC | SRCC | PLCC | SRCC | PLCC | SRCC | PLCC | SRCC | PLCC | SRCC |
| DACOM$_{SSIM}$ | 97.71 | 90.10 | 97.71 | 95.44 | 98.59 | 95.91 | 97.74 | 95.49 | 98.15 | 95.76 | 96.05 | 91.11 | 97.66 | 93.97 |
| DACOM$_{MS-SSIM}$ | 98.03 | 90.55 | 97.70 | 95.57 | 98.62 | 96.12 | 97.77 | 95.54 | 98.24 | 95.08 | 96.01 | 91.23 | 97.73 | 94.01 |
| DACOM$_{FSIM}$ | 97.65 | 90.83 | 97.75 | 95.13 | 98.70 | 96.01 | 97.70 | 95.37 | 97.88 | 94.90 | 94.79 | 90.18 | 97.41 | 93.74 |
| DACOM$_{DISTS}$ | 97.70 | 90.93 | 97.59 | 95.21 | 98.64 | 96.38 | 97.77 | 95.54 | 98.16 | 95.70 | 96.05 | 91.70 | 97.65 | 94.24 |

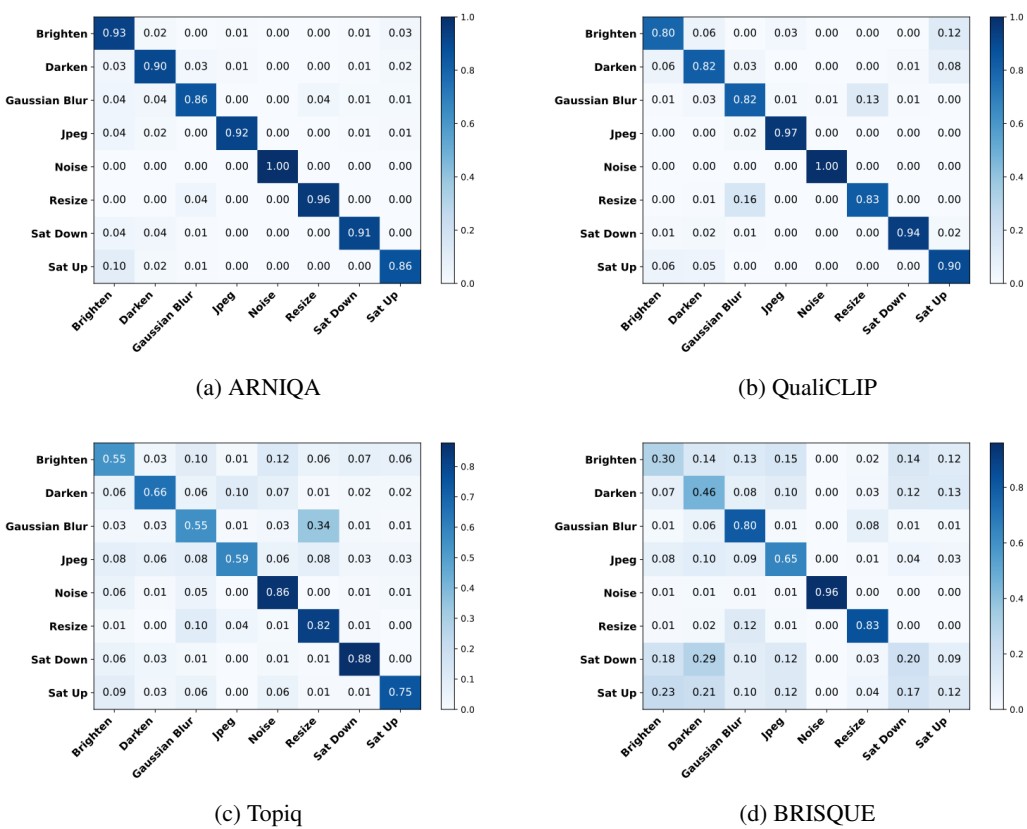

Figure 12: Confusion Matrix for Distortion Type Classification

# F IMPLEMENTATION DETAILS

For our distortion-aware confidence model(DACOM), we employed the Adam optimizer with a weight decay term. The batch size was set to 64, and the model was trained for 5 epochs with an initial learning rate of $1 \times 10^{-5}$. A linear warm-up strategy was applied during the first epoch, where the learning rate increased gradually from 10% of the initial value to the full rate. After the warm-up, a cosine annealing schedule was used, with the minimum learning rate set to 5% of the initial value. Model checkpoints are selected based on the highest SRCC achieved on the validation set. The input to the Confidence Model's Image Quality Encoder and Distortion-Type Encoder is of size 256×256 pixels, while the input to the Forensic Detector Encoder is processed according to the specific requirements of the corresponding detector.

# G DISTORTION-TYPE CONFIDENCE WITH DETECTION PERFORMANCE

Table 9: Correlation between Distortion-Type Confidence and Detection Performance (%).

| Method | CD | | ND | | NPR+ | | FreqNet+ | | SAFE+ | | C2P* | | Average | |
|---|---|---|---|---|---|---|---|---|---|---|---|---|---|---|
| | PLCC | SRCC | PLCC | SRCC | PLCC | SRCC | PLCC | SRCC | PLCC | SRCC | PLCC | SRCC | PLCC | SRCC |
| DACOM_{SSIM} | 93.25 | 83.82 | 86.49 | 86.18 | 90.83 | 85.80 | 90.67 | 86.76 | 98.23 | 88.95 | 71.65 | 75.94 | 88.52 | 84.58 |
| DACOM_{MS-SSIM} | 93.10 | 87.35 | 85.79 | 87.94 | 92.34 | 89.92 | 92.35 | 88.53 | 98.38 | 88.07 | 71.63 | 79.18 | **88.93** | **86.83** |
| DACOM_{FSIM} | 92.78 | 81.47 | 85.13 | 86.47 | 91.55 | 86.83 | 92.18 | 87.35 | 97.54 | 90.43 | 67.99 | 80.50 | 87.86 | 85.51 |
| DACOM_{DISTS} | 93.78 | 81.76 | 85.93 | 87.65 | 92.29 | 88.01 | 90.99 | 87.35 | 98.33 | 90.13 | 69.81 | 82.27 | 88.52 | 86.19 |

The regression performance of DACOM across all detectors is summarized in Table 8. We evaluate the relationship between the average confidence and accuracy of our proposed confidence model on datasets comprising various distortion types, including both seen and unseen distortions. As shown in the Table 9, we report the PLCC and SRCC correlation metrics between average confidence and

accuracy. Using MS-SSIM as the full-reference IQA method, the model achieves average PLCC and SRCC values of 88.93% and 86.83%, respectively. Experimental results demonstrate that the confidence model generalizes well across different distortion types, and its outputs exhibit a strong correlation with detection performance, effectively predicting the likelihood of correct detection.

## H    MULTI-DISTORTION FILTERING

As shown in Table 10 and Table 11, we present the filtering results under random multiple distortions for the Evaluation-dataset and Cross-dataset, respectively. This setting is particularly challenging, especially for Cross-dataset, as most compared methods show limited performance gains. While this is the case, our approach still manages to achieve competitive results in BAcc and EER across most screening ratios.

Table 10: Results on Evaluation-dataset Multi-Distortion Filtering

| Method | BAcc↑ | EER↓ | 0.05 | | 0.10 | | 0.15 | | 0.20 | | 0.30 | | 0.40 | |
|---|---|---|---|---|---|---|---|---|---|---|---|---|---|---|
| Topiq | 72.79 | 25.77 | 73.05 | 25.38 | 73.31 | 25.22 | 73.61 | 24.98 | 73.80 | 24.70 | 74.47 | 24.03 | 75.22 | 23.33 |
| ARNIQA | | | 72.94 | 25.70 | 73.03 | 25.66 | 73.11 | 25.62 | 73.34 | 25.56 | 73.65 | 25.57 | 73.84 | 25.35 |
| QualiCLIP | | | 73.22 | 25.24 | 73.54 | 24.88 | 74.08 | 24.37 | 74.55 | 23.98 | 75.35 | 23.25 | 76.12 | 22.66 |
| Logit Calib. | | | 73.63 | 25.50 | 74.45 | 25.43 | 75.01 | 25.37 | 75.45 | 25.41 | 75.92 | 25.15 | 76.46 | 24.69 |
| FR-SSIM | 72.79 | 25.77 | 73.33 | 25.17 | 73.82 | 24.76 | 74.48 | 24.02 | 75.00 | 23.50 | 76.03 | 22.59 | 77.15 | 21.60 |
| FR-MS-SSIM | | | 73.14 | 25.36 | 73.51 | 24.90 | 73.79 | 24.64 | 74.28 | 24.26 | 75.19 | 23.39 | 76.31 | 22.28 |
| FR-FSIM | | | 73.00 | 25.52 | 73.21 | 25.33 | 73.66 | 24.95 | 74.03 | 24.61 | 74.72 | 23.79 | 75.94 | 22.45 |
| FR-DISTS | | | 73.26 | 25.21 | 73.56 | 24.93 | 73.97 | 24.53 | 74.29 | 24.14 | 75.32 | 23.07 | 76.69 | 21.63 |
| DACOM$_{SSIM}$ | 72.79 | 25.77 | 73.78 | 24.61 | 74.69 | 23.54 | 75.70 | 22.63 | 76.70 | 21.70 | 78.02 | 20.24 | 78.72 | 19.09 |
| DACOM$_{MS\text{-}SSIM}$ | | | 73.77 | 24.55 | 74.61 | 23.62 | 75.76 | 22.60 | 76.73 | 21.64 | 78.18 | **20.11** | 78.83 | 19.01 |
| DACOM$_{FSIM}$ | | | 73.79 | 24.53 | 74.75 | **23.46** | 75.72 | 22.64 | 76.73 | 21.66 | **78.29** | **20.11** | 78.70 | 19.06 |
| DACOM$_{DISTS}$ | | | **73.87** | **24.51** | **74.83** | **23.46** | **75.91** | **22.42** | **76.85** | **21.56** | 78.26 | 20.13 | **78.95** | **18.76** |

Table 11: Results on Cross-dataset Multi-Distortion Filtering

| Method | BAcc↑ | EER↓ | 0.05 | | 0.10 | | 0.15 | | 0.20 | | 0.30 | | 0.40 | |
|---|---|---|---|---|---|---|---|---|---|---|---|---|---|---|
| Topiq | 60.64 | 36.20 | 60.86 | 35.95 | 61.06 | 35.68 | 61.31 | 35.40 | 61.55 | 35.06 | 61.94 | 34.62 | 62.47 | 33.84 |
| ARNIQA | | | 60.54 | 36.29 | 60.46 | 36.35 | 60.41 | 36.50 | 60.44 | 36.44 | 60.32 | 36.64 | 60.10 | 36.74 |
| QualiCLIP | | | 61.00 | 35.76 | 61.37 | 35.21 | 61.70 | 34.83 | 61.98 | 34.49 | 62.54 | 33.82 | 63.06 | 33.42 |
| Logit Calib. | | | 60.68 | 36.37 | 60.69 | 36.68 | 60.65 | 37.08 | 60.71 | 37.47 | 60.81 | 37.80 | 61.12 | 38.28 |
| FR-SSIM | 60.64 | 36.20 | 60.95 | 35.85 | 61.27 | 35.53 | 61.61 | 35.09 | 61.96 | 34.76 | 62.72 | 33.98 | 63.46 | 33.41 |
| FR-MS-SSIM | | | 60.82 | 36.06 | 61.11 | 35.77 | 61.42 | 35.46 | 61.77 | 34.96 | 62.48 | 34.17 | 63.20 | 33.45 |
| FR-FSIM | | | 60.83 | 36.00 | 61.04 | 35.84 | 61.29 | 35.11 | 61.60 | 35.21 | 62.30 | 34.45 | 63.14 | 33.64 |
| FR-DISTS | | | **61.03** | 35.72 | **61.44** | 35.24 | **61.83** | 34.70 | 62.22 | 34.34 | 63.08 | 33.36 | 63.92 | 32.54 |
| DACOM$_{SSIM}$ | 60.64 | 36.20 | 60.93 | 35.71 | 61.30 | 35.06 | 61.73 | 34.50 | 62.21 | 33.96 | 63.15 | 33.06 | 64.02 | 32.46 |
| DACOM$_{MS\text{-}SSIM}$ | | | 60.95 | 35.65 | 61.35 | **35.00** | 61.79 | 34.43 | 62.28 | 33.86 | 63.20 | 33.02 | 64.05 | 32.29 |
| DACOM$_{FSIM}$ | | | 60.96 | 35.71 | 61.35 | 35.07 | 61.77 | 34.54 | **62.36** | 33.89 | **63.35** | **32.97** | 64.12 | 32.33 |
| DACOM$_{DISTS}$ | | | 60.97 | **35.64** | 61.34 | 35.01 | 61.77 | **34.44** | 62.25 | **33.85** | 63.22 | 32.98 | **64.19** | **32.23** |

## I    APPLICABILITY OF DACOM ACROSS DIFFERENT DATA DOMAIN

We retrained our detector on the SD14 subset of GenImage (Zhu et al., 2023), following the same experimental protocol and hyperparameter configuration. During training, `FreqNet` and `SAFE` failed to converge under the required augmentation pipeline—likely due to sensitivity to the high-frequency artifacts and distributional characteristics of diffusion-generated images. Therefore, we report results only for `CD`, `ND`, `NPR`, and `C2P`. Similarly, `C2P` uses the SD14-pretrained weights. All models, including DACOM$_{DISTS}$, are trained on SD14, and baselines use temperature-scaled logit calibration. The distortion types and data volume used during training and evaluation are fully consistent with the original paper's configuration. We compare our proposed method (using

$\text{DACOM}_{\text{DISTS}}$ as an example) against the outputs of the calibrated detectors. As shown in the Table 12, $\text{DACOM}_{\text{DISTS}}$ consistently outperforms baseline calibration methods across both seen and unseen distortions, delivering superior average and worst-case performance.

Table 12: Multi-detector fusion performance on SD14 (%; Acc↑ / AP↑)

| Method | Seen distortion | | Unseen distortion | | Average | | Worst | |
|---|---|---|---|---|---|---|---|---|
| | Acc | AP | Acc | AP | Acc | AP | Acc | AP |
| Logit Calib. | 91.85 | 99.00 | 94.64 | 98.67 | 93.59 | 98.79 | 72.70 | 90.14 |
| $\text{DACOM}_{\text{DISTS}}$ | **97.70** | **99.70** | **96.10** | **98.95** | **96.70** | **99.23** | **81.60** | **93.38** |

Moreover, it facilitates a more favorable precision–coverage trade-off for confidence-based filtering on SD14, and generalizes well to cross-dataset evaluation on SD15. As shown in Tables 13–16, DACOM consistently surpasses baseline in the filtering task across every setting—including in-domain and cross-domain datasets, as well as both single- and multi-distortion scenarios.

Table 13: Single-Distortion filtering (%) in SD14

| Method | Distortion | | Filtering Proportion | | | | | | | |
|---|---|---|---|---|---|---|---|---|---|---|
| | BAcc ↑ | EER ↓ | 0.10 | | 0.20 | | 0.30 | | 0.40 | |
| Logit Calib. | 87.77 | 11.82 | 89.88 | 11.46 | 91.11 | 10.78 | 92.62 | 9.72 | 94.20 | 8.38 |
| $\text{DACOM}_{\text{DISTS}}$ | 87.77 | 11.82 | **90.31** | **7.97** | **92.68** | **5.22** | **94.29** | **4.02** | **95.50** | **2.92** |

Table 14: Single-Distortion filtering (%) in SD15

| Method | Distortion | | Filtering Proportion | | | | | | | |
|---|---|---|---|---|---|---|---|---|---|---|
| | BAcc ↑ | EER ↓ | 0.10 | | 0.20 | | 0.30 | | 0.40 | |
| Logit Calib. | 87.74 | 11.60 | 89.78 | 11.31 | 91.18 | 10.81 | 92.62 | 9.93 | 94.06 | 8.60 |
| $\text{DACOM}_{\text{DISTS}}$ | 87.74 | 11.60 | **90.28** | **7.77** | **92.69** | **5.18** | **94.34** | **3.89** | **95.55** | **2.95** |

Table 15: Multi-Distortion filtering (%) in SD14

| Method | Distortion | | Filtering Proportion | | | | | | | |
|---|---|---|---|---|---|---|---|---|---|---|
| | BAcc ↑ | EER ↓ | 0.10 | | 0.20 | | 0.30 | | 0.40 | |
| Logit Calib. | 76.81 | 21.76 | 77.77 | 22.66 | 78.19 | 23.63 | 78.44 | 24.42 | 78.62 | 24.75 |
| $\text{DACOM}_{\text{DISTS}}$ | 76.81 | 21.76 | **78.69** | **18.93** | **81.01** | **15.80** | **83.00** | **13.90** | **84.51** | **12.48** |

Table 16: Multi-Distortion filtering (%) in SD15

| Method | Distortion | | Filtering Proportion | | | | | | | |
|---|---|---|---|---|---|---|---|---|---|---|
| | BAcc ↑ | EER ↓ | 0.10 | | 0.20 | | 0.30 | | 0.40 | |
| Logit Calib. | 76.21 | 22.12 | 77.36 | 23.12 | 77.74 | 24.17 | 77.97 | 25.00 | 77.99 | 25.61 |
| $\text{DACOM}_{\text{DISTS}}$ | 76.21 | 22.12 | **78.01** | **19.52** | **80.52** | **16.32** | **82.54** | **14.17** | **83.95** | **12.76** |

## J ABLATION RESULTS

As shown in Table 17, our ablation study shows that adding each proposed module leads to a substantial increase in both PLCC and SRCC, demonstrating their individual necessity and effectiveness.

Table 18 reports the impact of different mappings (e.g., linear, power-law, logarithmic) on the final routing performance. The results show that the simple linear mapping (*Linear*) achieves the best performance. This observation reveals an intriguing empirical regularity: in a statistical sense, the detector's detectability strength is approximately linearly related to the *degree of deviation from random guessing*. In contrast, overly nonlinear transformations may introduce additional noise or distort the underlying distributional characteristics.

Table 17: Ablation studies on the effectiveness of each module (%). Here, $\phi_{\text{IQ}}^{(Q)}$ and $\phi_{\text{IQ}}^{(T)}$ denote the use of QualiCLIP and Topiq as the Image Quality Encoder, respectively.

| $\phi_{\text{M}}$ | $\phi_{\text{IQ}}^{(T)}$ | $\phi_{\text{IQ}}^{(Q)}$ | $\phi_{\text{DT}}$ | CD | | ND | | NPR+ | | FreqNet+ | | SAFE+ | | C2P* | | Average | |
|---|---|---|---|---|---|---|---|---|---|---|---|---|---|---|---|---|---|
| | | | | PLCC | SRCC | PLCC | SRCC | PLCC | SRCC | PLCC | SRCC | PLCC | SRCC | PLCC | SRCC | PLCC | SRCC |
| ✓ | | | | 93.40 | 78.22 | 93.76 | 87.91 | 97.23 | 94.04 | 90.86 | 86.50 | 94.21 | 91.33 | 86.86 | 81.88 | 92.72 | 86.65 |
| ✓ | ✓ | | | 95.14 | 81.41 | 95.03 | 90.34 | 97.80 | 94.29 | 94.04 | 90.16 | 95.61 | 92.74 | 88.18 | 82.63 | 94.30 | 88.60 |
| ✓ | | ✓ | | 96.49 | 85.92 | 96.48 | 92.99 | 98.33 | 95.12 | 96.56 | 93.23 | 96.78 | 94.12 | 91.99 | 87.01 | 96.11 | 91.40 |
| ✓ | | ✓ | ✓ | **97.71** | **90.10** | **97.71** | **95.44** | **98.58** | **95.91** | **97.74** | **95.49** | **98.15** | **95.76** | **96.05** | **91.11** | **97.66** | **93.97** |

Table 18: Ablation on label transformation functions for DACOM training. Performance (Acc↑ / AP↑) of multi-detector fusion under different label mappings.

| Label functions | $\alpha$ | ID distortion | OOD distortion | Average | Worst |
|---|---|---|---|---|---|
| Linear (Ours) | – | **95.37/99.47** | **92.84/98.33** | **93.79/98.76** | **84.90/93.16** |
| $y_{\text{pow}} = (y)^{\alpha}$ | 0.5 | 95.20/99.41 | 92.46/97.78 | 93.49/98.39 | 82.80/90.66 |
| $y_{\text{pow}} = (y)^{\alpha}$ | 2.0 | 95.35/**99.47** | 92.65/98.19 | 93.66/98.67 | 84.00/92.15 |
| $y_{\text{log}} = \frac{\log(1+\alpha y_{\text{lin}})}{\log(1+\alpha)}$ | 3.0 | 95.13/99.38 | 92.29/97.58 | 93.36/98.26 | 82.80/90.46 |
| $y_{\text{exp}} = \frac{1-e^{-\alpha y_{\text{lin}}}}{1-e^{-\alpha}}$ | 3.0 | 94.72/99.12 | 92.14/97.54 | 93.11/98.14 | 82.10/90.16 |

## K    REPRODUCIBILITY STATEMENT

We have made every effort to ensure the reproducibility of our work. All datasets used in this paper are publicly available and the exact training/testing splits and preprocessing steps are documented in Appendix D. Our proposed Distortion-Aware Confidence Model (DACOM) is described in detail, including its motivation in Section 3, methodological steps in Section 4, training procedures in Section 4.4, and hyperparameters and implementation details in Section 5.1 and Appendix F. To further facilitate reproduction, we will release our source code, pretrained models, and data-processing scripts upon publication.

## L    THE USE OF LARGE LANGUAGE MODELS (LLMS)

During the preparation of this manuscript, Large Language Models (LLMs), specifically GPT-5, were employed as versatile writing and research assistants. Their primary contributions included:

- **Refining and Polishing Language:** Improving clarity, conciseness, and grammatical accuracy of the text, with particular attention to academic English style and phrasing.
- **Formatting LaTeX Code:** Supporting the generation and debugging of LaTeX code for tables, figures, and mathematical equations, thereby ensuring professional presentation and consistent formatting.

It is important to note that all core research ideas, experimental design, implementation, and interpretation of results were independently conceived and conducted by the human authors. The LLM was used solely as a tool to enhance the quality and readability of the manuscript's presentation, without contributing to the original scientific findings.

