# OpenReview forum: "Enabling Your Forensic Detector Know ​How Well​ It Performs on Distorted Samples"
_ICLR.cc/2026/Conference — ICLR 2026 Poster_

### Official Review · Reviewer_2NT5 · 2025-10-24

**Soundness:** 4
**Presentation:** 3
**Contribution:** 3
**Rating:** 8
**Confidence:** 4

**Summary:**

This paper proposes DACOM, a neural regressor that predicts the probability that a given forensic detector will correctly classify a (possibly distorted) image. The key insight is that FR-IQA scores correlate monotonically with detector accuracy conditioned on distortion type. Training labels are obtained by bucketing FR-IQA scores per distortion type and mapping bin-wise balanced accuracies to [0,1]. At inference DACOM uses detector features, NR-IQA features and a distortion-type embedding to estimate sample-level confidence without references. The score enables “selective abstention’’ and “top-1 routing’’ among detectors and improves overall accuracy on several benchmark.

**Strengths:**

1. The paper introduces a novel framework that tackles reliability estimation, which is orthogonal and complementary to existing works.
2. It provides a principled, detector-conditioned confidence definition and a practical solution that avoids requiring reference images at test time.
3. Extensive experimental validation on diverse detectors and a broad spectrum of distortions is conducted. Ablations clearly show each component’s contribution.
4. The paper is well written and easy to follow.

**Weaknesses:**

1. Evaluations rely mainly on Balanced Accuracy and EER. In practice, real and fake samples are imbalanced. Would precision-recall–based measures (e.g., AUC-PR, F1) change the conclusions?
2. The inference pipeline runs QualiCLIP, ARNIQA and the detector for every input may be costly on edge devices. A detailed analysis of timing/FLOPs is missing.
3. While abstention can improve safety, it also off-loads decisions to human operators and potentially offers adversaries a mechanism to trigger systematic abstention. The paper lacks discussion of these risks and possible mitigations.

**Questions:**

1. How would the results change if Balanced Accuracy is replaced with AUC-PR or F1 measures?
2. Please report the inference overhead of DACOM and compare it to baselines.
3. Please add more discussions about ethical aspects.
4. Minor issue: Text in Fig. 2 and Fig.6 is tiny, please enlarge.

---

> ### Author Response · Authors · 2025-11-20
> **Response to Reviewer 2NT5**
>
> 1. **Regarding Weaknesses (1) and Questions (1):** When replacing balanced accuracy with AUC-PR or F1, the evaluation results show the same trend—as the filtering proportion increases, the corresponding detection performance also improves. We take $DACOM_{DISTS}$ as an example and evaluate the post-filtering results on both the Evaluation-dataset and the Cross-dataset under single-distortion and multi-distortion settings. We report two metrics: F1 and AUC-PR. Here, “Filt-Prop.” denotes the filtering proportion, “0.00” indicates a proportion of 0.00, and values such as “84.30/89.93” correspond to “F1 / AUC-PR,” respectively. The results are as follows:
>
>     | Dataset              | Distortion type   | Filt-Prop. 0.00          | Filt-Prop. 0.10          | Filt-Prop. 0.20          | Filt-Prop. 0.30          | Filt-Prop. 0.40          |
>     |---------------------|--------------|---------------|---------------|---------------|---------------|---------------|
>     | Evaluation-dataset        |  Single  | 84.30/89.93   | 87.65/92.48   | 89.94/95.20    | 91.56/97.41    | 93.38/98.88    |
>     | Evaluation-dataset | Multiple  | 72.20/80.23   | 74.46/82.08   | 76.46/84.53    | 77.72/86.57    | 78.31/87.38    |
>     | Cross-dataset        |  Single  | 68.75/79.67   | 70.83/82.17   | 72.74/85.09    | 74.77/86.83    | 76.44/88.95    |
>     | Cross-dataset | Multiple  | 57.53/69.29   | 58.55/70.41   | 59.70/72.21    | 60.74/73.96    | 61.73/75.02    |
>
>
> 2. **Regarding Weaknesses (2) and Questions (2):** We thank the reviewer for raising the important concern regarding the computational cost of our inference pipeline on edge devices. In response, we have conducted a comprehensive analysis of both theoretical computation (FLOPs) and actual inference latency (timing) across all modules. It is worth noting that across all evaluated detectors, the DACOM module maintains a consistent architectural design; only the input projection layer is adjusted to accommodate the varying feature dimensions output by each detector. Consequently, the computational footprint (parameters, FLOPs, and latency) of DACOM remains largely invariant.
>
>     As shown in the example result for C2P with input size 512×512.
>     The table shows DACOM adds only 9.6% params and 16.5% FLOPs, with ~54% latency contribution on L40 — indicating moderate overhead. With optimization (e.g., quantization), the pipeline remains viable for edge deployment.
>
>
>     | Metric                | Detector (C2P) | DACOM | Total Pipeline | DACOM vs Detector (%) |
>     |-----------------------|---------------------|----------------------------|----------------|------------------------|
>     | Parameters (M)    | 303.97              | 29.26                      | —              | 9.63%                 |
>     | FLOPs (GFLOPs)    | 81.08               | 13.37                      | 94.44          | 16.49%                |
>     | Avg. Latency (ms) | 10.24               | 8.99                       | 16.73          | 53.8% (of total)      |
>     | FPS (L40 GPU)     | —                   | —                          | 59.8           | —                     |
>     | Time Contribution | 61.2%               | 53.8%                      | 100%           | —                     |
>
>     > *Note: Time contributions sum to >100% because they are measured independently; actual execution may involve overlapping computation. Input size: 512×512, device: NVIDIA L40.*
>
> 3. **Regarding Weaknesses (3) and Questions (3):** We appreciate the reviewer’s insight into the ethical implications of abstention. While acknowledging risks (e.g., responsibility shift, adversarial abuse), we note that abstention is standard practice in safety-critical domains:
>
>     1. Facial recognition systems routinely reject low-confidence matches to prevent false positives;
>     2. Medical diagnostic AI tools flag ambiguous cases (e.g., uncertain tumor classifications) for expert radiologist review;
>     3. Autonomous vehicles hand over control to human drivers when sensor or decision uncertainty exceeds predefined thresholds.
>
>     In each case, **abstention functions as a deliberate safety mechanism — not a system failure — designed to preserve reliability in high-stakes environments**. We recognize that our current manuscript does not sufficiently address these ethical and operational dimensions — an important gap we intend to fill. In the revised version, we will add a dedicated subsection to address these concerns, drawing parallels from real-world deployments in industry and prior work on safe AI deployment.
>
> 4. Issues such as small text and plots that affect readability will be addressed in our subsequent revised version. Thank you for pointing these out.

---

### Official Review · Reviewer_5Ewd · 2025-10-31

**Soundness:** 4
**Presentation:** 4
**Contribution:** 4
**Rating:** 6
**Confidence:** 5

**Summary:**

This paper addresses the problem that forensic detectors for AI-generated images produce predictions without indicating reliability when test images undergo various distortions such as compression and noise. The authors propose DACOM (Distortion-Aware Confidence Model), which uses full-reference image quality assessment metrics to label training data with detectability scores during training, then learns to predict sample-level confidence using detector features, no-reference image quality descriptors, and distortion-type information at inference time.

**Strengths:**

1. The paper's motivation and perspective are well-founded and novel. Few existing works approach AI-generated image detection from the lens of image quality assessment to evaluate detector reliability under distortions. This angle provides a fresh and meaningful contribution to the forensics community.
2. The paper is well-organized with clear logical flow. The authors systematically progress from problem identification (Section 3 analysis) to method design (Section 4), making the work easy to follow. The empirical analysis establishing the correlation between FR-IQA scores and detection accuracy provides solid justification for the proposed approach.
3. The experimental evaluation is comprehensive and thorough. The authors provide extensive ablation studies (Section 5.5), test on multiple distortion types (both seen and unseen), evaluate across different datasets (Evaluation-dataset and Cross-dataset), and include detailed supplementary results in the appendix, all of which substantiate their claims with adequate evidence.

**Weaknesses:**

1. Concerns regarding the use of detector performance for labeling in Stage A. The authors use the detector's balanced accuracy on each bin to generate detectability labels (Equation 3-4). This raises several concerns: (a) If FR-IQA scores already exhibit monotonic correlation with detection performance (as shown in Section 3), why is the additional step of computing detector accuracy necessary? Could the FR-IQA scores themselves serve as supervision? (b) More critically, this design may limit generalizability—if training data is labeled using Detector A's performance, will DACOM trained on this data generalize well to Detector B? This detector-specific labeling could hinder practical deployment across different detection models. (c) The requirement to evaluate detector performance on large distorted datasets during Stage A significantly increases the computational cost of the training pipeline.
2. Limited discussion on robustness training and its interaction with the observed monotonicity. The authors address distortion robustness by applying light data augmentation (10% JPEG compression and blur) during detector training. However: (a) Only two distortion types are used for augmentation, which seems insufficient given the diversity of real-world distortions. (b) It remains unclear whether the monotonic relationship between FR-IQA and detection accuracy (Section 3) still holds when detectors are trained with more aggressive data augmentation strategies. If extensive augmentation flattens the performance curve across distortion levels, would DACOM's premise still be valid? This interaction between robustness training and the proposed method deserves further investigation.
3. Limited training data scale may restrict generalization. The model is trained on only 2,500 base images from a single dataset (ProGAN subset). While distortion augmentation expands this to 200K+ samples, the underlying content diversity remains limited. This could lead to: (a) Overfitting to the specific visual patterns in these 2,500 images. (b) Poor generalization to different content types, as evidenced by the noticeable performance drop in Cross-dataset evaluation (Table 4). Experimenting with larger and more diverse training sets would strengthen the claims about generalizability.
4. The proposed applications have practical limitations that merit further exploration. While the paper demonstrates two uses of DACOM—selective filtering and multi-detector routing—both have notable drawbacks: (a) Selective abstention necessarily reduces coverage, which may be unacceptable in applications like content moderation where all samples must be processed. (b) Multi-detector routing requires maintaining and running multiple detectors (6× computational cost in experiments), which may be prohibitively expensive for real-time systems. I suggest the authors explore alternative applications that leverage DACOM more seamlessly, such as incorporating confidence-aware calibration directly into a single detector's training or inference process, or using confidence scores to dynamically adjust decision thresholds rather than completely abstaining from prediction.

Overall assessment: Despite these concerns, I view this as a valuable contribution that introduces a novel perspective on detector reliability. If the authors can adequately address the above concerns—I would be inclined to raise my score.

**Questions:**

1. The formula y = 2×|BAcc - 0.5| assumes equal difficulty in improving accuracy across the entire range (e.g., 50%→75% vs. 75%→100%). However, achieving near-perfect accuracy is typically much harder than reaching moderate levels, suggesting a non-linear relationship.
Have you experimented with non-linear transformations (e.g., y = (2×|BAcc - 0.5|)^α with α > 1, or logarithmic scaling) that better reflect the diminishing returns at higher accuracy? Alternatively, why not use BAcc directly as labels without transformation? An ablation study comparing different label functions would clarify whether this linear design is optimal or just a convenient choice.

2. You train four DACOM variants with different FR-IQA metrics (Table 6) and all perform similarly. Does this mean the choice of FR-IQA is not critical? If so, why present four variants instead of selecting one? What guidance do you offer to practitioners on choosing the FR-IQA metric?

3. What is the parameter count and FLOPs of DACOM? Since it must run alongside the detector, efficiency matters. How does DACOM's overhead compare to the detector itself (e.g., DACOM adds X% latency)?

---

> ### Author Response · Authors · 2025-11-20
> **Response to Reviewer 5Ewd (Part 1)**
>
> 1. 1. **Regarding Weaknesses (1)(a):** While FR-IQA scores exhibit strong monotonic correlation with detection performance, they do not encode distortion-specific impacts on detectors. As illustrated in Figs. 2 and 6, different distortion types can yield similar FR-IQA scores yet produce substantially different detection behaviors (e.g., blur vs. JPEG compression at the same SSIM level). This mismatch indicates that FR-IQA scores lack the granularity required for distortion-aware performance modeling. Therefore, they are insufficient as direct supervision labels.
>
>    2. **Regarding Weaknesses (1)(b):** We acknowledge that our detector-specific labeling limits cross-detector generalization. However, this design aligns with our core objective: accurately predicting the performance of a given detector under distortion, not learning a universal proxy. Labels derived from the target detector explicitly encode its unique sensitivity patterns—such as how it fails under noise versus geometric distortion—which would be lost in an averaged or model-agnostic label. While this reduces immediate deployability across arbitrary detectors, it enables precise, per-detector calibration. We agree that learning a more transferable representation—e.g., via detector-agnostic distortion modeling—is a compelling direction for future work.
>
>    3. **Regarding Weaknesses (1)(c):** You are right — we do evaluate each distortion dataset prior to training to generate labels, which incurs additional computational overhead. We acknowledge this as a practical limitation and agree that reducing this dependency is an important direction for future work.
>
> 2. **Regarding Weaknesses (2):** We appreciate this insightful comment. We address the two concerns as follows:
>
>     1. **On the choice of only two distortion types for augmentation:**
>         Our robustness training follows the protocol in [1], using JPEG compression and Gaussian blur — two canonical and widely adopted distortions that effectively simulate realistic degradation patterns commonly observed in real-world imaging conditions.
>
>         While more diverse augmentations could be considered, we intentionally limit their number based on empirical observation: excessive or overly aggressive data augmentation suppresses discriminative feature learning, leading to poor convergence and unstable performance. In such cases, the detector fails to learn reliable discriminative features, making its accuracy an unreliable signal for modeling degradation trends. Our choice thus balances robustness exposure with model trainability.
>     2. **On the validity of monotonicity under stronger augmentation:**
>         To verify that the monotonic relationship between FR-IQA scores and detection performance is not an artifact of weak regularization, we trained two variants of the CD detector: one with robustness augmentation (JPEG compression and Gaussian blur) and one without.
>         For analysis, we used DISTS as the FR-IQA metric to group distorted samples into quality bins and evaluate detection performance within each bin. As shown in Table 1, the performance degradation trend remains strictly monotonic across all distortion types — regardless of whether augmentation was applied.
>
>         Table 1 : Effect of Robustness Augmentation on the Monotonic Relationship Between FR-IQA Scores and Detection Accuracy.
>         Each cell reports the balanced accuracy of the model with augmentation and without augmentation.
>         | Distortion Type   | [1.00, 0.95)     | [0.95, 0.90)     | [0.90, 0.85)     | [0.85, 0.80)     | [0.80, 0.75)     |
>         |-------------------|------------------|------------------|------------------|------------------|------------------|
>         | JPEG              | 0.97 / 0.67      | 0.96 / 0.50      | 0.94 / 0.50      | 0.89 / 0.50      | 0.87 / 0.50      |
>         | Blur              | 0.99 / 0.99      | 0.99 / 0.99      | 0.98 / 0.92      | 0.98 / 0.76      | 0.98 / 0.59      |
>         | Noise             | 0.95 / 0.88      | 0.80 / 0.58      | 0.61 / 0.50      | 0.53 / 0.50      | 0.51 / 0.50      |
>         | Color warming     | 0.88 / 0.94      | 0.80 / 0.90      | 0.71 / 0.85      | 0.64 / 0.78      | 0.62 / 0.72      |
>         | Brighten          | 0.98 / 0.98      | 0.96 / 0.96      | 0.93 / 0.95      | 0.88 / 0.90      | 0.85 / 0.89      |
>
>
>
>         We agree that exploring extreme augmentation regimes is valuable. However, when augmentation intensity prevents convergence or erases discriminative capability, performance becomes noisy and non-monotonic—undermining any meaningful performance prediction. Thus, DACOM operates within a practical regime where detectors remain functional yet sensitive to distortion severity.
>
> [1]Wang, S. Y., Wang, O., Zhang, R., Owens, A., & Efros, A. A. (2020). CNN-generated images are surprisingly easy to spot... for now. In Proceedings of the IEEE/CVF conference on computer vision and pattern recognition (pp. 8695-8704).

---

> ### Author Response · Authors · 2025-11-20
> **Response to Reviewer 5Ewd (Part 2)**
>
> 3. **Regarding Weaknesses (3):** We agree that training on only 2,500 base images from the ProGAN subset limits content diversity. However, we would like to clarify that the proposed DACOM framework has also been trained and validated on a substantially different dataset (SD14), and we observe the same distortion–performance relationships and monotonic trends (for detailed results, please refer to our response to Review iFaB), consistent with our conclusions in the ProGAN setting. This cross-dataset consistency indicates that our findings are not tied to the content characteristics of the 2,500 ProGAN images, and the core effectiveness of DACOM does not rely on a particular dataset.
>
>     That said, we acknowledge the reviewer’s point: the ProGAN-based experiment setup is intentionally designed as a controlled environment to isolate the effect of distortions and analyze the FR-IQA–performance relationship without the confounding influence of heterogeneous content. The performance drop observed in cross-dataset evaluation indeed reflects the limited content diversity in this controlled setup.
>
>     While we have not yet scaled to larger datasets due to computational constraints, we recognize this as a critical next step. The ProGAN-based training serves as a controlled testbed to validate DACOM’s core premise; we will extend it to more diverse sources in future work.
>     More fundamentally, long-term robustness requires moving beyond data-scale scaling alone. Architectural or regularization strategies that encourage learning of distortion-intrinsic features — rather than memorizing content-specific artifacts — can complement data diversity. We view this as a complementary direction, not a replacement, and thank the reviewer for highlighting this essential challenge.
>
> 4. **Regarding Weaknesses (4)(a):** We agree that abstention reduces coverage and is therefore not suitable for settings where every sample must be processed (e.g., content moderation). However, **abstention remains highly valuable in high-risk or safety-critical scenarios**, where rejecting low-confidence inputs can prevent harmful false positives/negatives — which is precisely the class of applications DACOM targets.
> That said, abstention is only one way to operationalize DACOM’s confidence estimates. For mandatory-coverage applications, DACOM can be used in a non-abstention mode, where its scores drive adaptive decision thresholds instead of rejection. This preserves 100% coverage while still improving calibration. We consider this a **complementary deployment option**, not a replacement for abstention.
>
>     **Regarding Weaknesses (4)(b):** We acknowledge that multi-detector routing introduces computational overhead. Its purpose in our work is to **highlight detector-specific distortion sensitivity**, which cannot be captured if all detectors are collapsed into a single model. This design is therefore useful for **analyzing robustness differences across architectures**, especially in forensic settings.
>     At the same time, we agree with the reviewer that integrating DACOM’s confidence signals directly into a single-detector pipeline—via adaptive thresholding, confidence-aware inference rules, or uncertainty-aware training losses—offers a more scalable alternative for real-time systems. We view multi-detector routing as a diagnostic and analytical tool, and these single-detector extensions as a natural and important direction for deployment-oriented use cases.

---

> ### Author Response · Authors · 2025-11-20
> **Response to Reviewer 5Ewd (Part 3)**
>
> 5. **Regarding Questions (1):** We thank the reviewer for pointing out this important detail. Upon reviewing our manuscript, we identified a minor inconsistency between the presented formula and our actual implementation.
> In the original submission, we wrote the label transformation as $ y = 2|BAcc - 0.5| $. However, in practice, we use:
>     $$
>     y = 2 \cdot \max(BAcc - 0.5, 0)
>     $$
>     This one-sided mapping reflects our design principle: **detectability strength should only be measured when performance exceeds chance level**. Our goal is to model detectability strength above chance, not absolute accuracy. Raw BAcc ∈ [0,1] mixes two regimes: values below 0.5 indicate random or inverted decisions, which are directionally ambiguous and incomparable across detectors. Thus, we treat BAcc < 0.5 as “no reliable signal.” We then map the informative interval [0.5,1] to [0,1] using y = 2×max(BAcc - 0.5, 0), which on meaningful variations in detectability strength and avoids noisy gradients from uninformative cases.
>
>     We have added ablation experiments on different label functions(Table 2), comparing various transformation forms—including power, exponential, and logarithmic functions—against our original linear transformation.
>
>     Table 2:  Ablation on label transformation functions for DACOM training. Performance (Acc↑ / AP↑) of multi-detector fusion under different label mappings.
>
>     | Label functions          | $\alpha$ | ID distortion | OOD distortion | Average    | Worst      |
>     |-----------------|---------------|---------------|----------------|------------|------------|
>     | Linear(Ours)    | -    | **95.37**/**99.47**   | **92.84/98.33**    | **93.79/98.76** | **84.90/93.16** |
>     | $ y_{\text{pow}} = (y)^{\alpha} $    | 0.5    | 95.20/99.41   | 92.46/97.78    | 93.49/98.39 | 82.80/90.66 |
>     | $ y_{\text{pow}} = (y)^{\alpha} $    | 2.0    | 95.35/**99.47**   | 92.65/98.19    | 93.66/98.67 | 84.00/92.15 |
>     | $ y_{\log} = \frac{\log(1 + \alpha y)}{\log (1 + \alpha)} $      | 3.0    | 95.13/99.38   | 92.29/97.58    | 93.36/98.26 | 82.80/90.46 |
>     | $ y_{\exp} = \frac{1 - e^{-\alpha y}}{1 - e^{-\alpha}} $      | 3.0    | 94.72/99.12   | 92.14/97.54    | 93.11/98.14 | 82.10/90.16 |
>
>     The linear formulation consistently achieves the best or tied-best performance across all metrics, demonstrating that detectability strength scales approximately linearly with deviation from chance. Non-linear transformations either compress mid-range signals or amplify noise near chance, degrading stability and generalization — confirming that our linear design is not merely convenient, but empirically optimal.
>
> 6. **Regarding Questions (2):** The framework remains effective across different FR-IQA designs, indicating that our method is not tied to any particular metric. We report all variants to demonstrate that DACOM is robust to the choice of FR-IQA metric. The four FR-IQA variants behave similarly because DACOM does not rely on the absolute FR-IQA values; it only requires a monotonic, distortion-aware signal to partition the data into performance-aligned intervals. All four FR-IQA metrics provide such a monotonic degradation trend, so the specific metric is not critical to the framework.
>
>
>     For practitioners, we recommend DISTS because it consistently achieves slightly better evaluation performance in our experiments, and it is also a more recent FR-IQA method with improved perceptual modeling compared to earlier metrics.
>
> 7. **Regarding Questions (3):** DACOM’s parameter count and FLOPs are largely detector-agnostic; only the projector’s input dimension adapts to the detector’s feature space. As shown in Table 2 for the C2P detector (512×512 input on an NVIDIA L40 GPU—a representative deployment setting), DACOM adds 29.26M parameters (9.6% of detector size) and 13.37 GFLOPs (16.5% increase). In terms of latency, DACOM introduces 8.99 ms, increasing the detector’s baseline (10.24 ms) by 87.8%, and accounts for 53.8% of total pipeline latency. While non-negligible, this overhead is moderate compared to the accuracy and reliability gains in distortion-aware prediction.
>
>     Table 3: Efficiency breakdown for C2P + DACOM (512×512 on L40)
>
>     | Metric                | Detector (C2P) | DACOM | Total Pipeline | DACOM vs Detector (%) |
>     |-----------------------|---------------------|----------------------------|----------------|------------------------|
>     | Parameters (M)    | 303.97              | 29.26                      | —              | 9.63%                 |
>     | FLOPs (GFLOPs)    | 81.08               | 13.37                      | 94.44          | 16.49%                |
>     | Avg. Latency (ms) | 10.24               | 8.99                       | 16.73          | 53.8% (of total)      |

---

> > ### Author Response · Authors · 2025-11-28
> > **Response to Reviewer 5Ewd**
> >
> > We appreciate the improved score you have given to our work, and we also thank you for all your comments. We will incorporate these suggestions to further refine and improve our work.

---

### Official Review · Reviewer_DcA6 · 2025-10-31

**Soundness:** 2
**Presentation:** 3
**Contribution:** 3
**Rating:** 6
**Confidence:** 5

**Summary:**

Conventionally, fake image detectors assign a real/fake label. However, detecting these fake images in the wild involves dealing with post-processing operations modify the original signal and hence affect their detection. The authors try to develop an effective method through which one can also predict the confidence of the detector. This is especially challenging since, neural networks are not well-calibrated. In order to do this, the authors leverage image quality to measure the confidence. The change in image quality, depends on the reference image, which is conventionally not available during test time, in order to predict this, the authors train the neural network to do so. Experiments show improved calibration and the authors also show how this can be used in effectively detecting fake images.

**Strengths:**

1. The problem of uncertainty estimation in fake image detection is both interesting as well as practically relevant. It is also an understudied problem.
2. The focus of the study on distortions also makes the paper practically relevant.
3. Multi-Detector Routing and Confidence-Based filtering are good use cases for the method.

**Weaknesses:**

1. The experiment reported in section 3 would benefit from the inclusion of more details. For instance, what is the data used, what is the amount to which each post-processing operation is applied, etc.
2. My main concern comes from the fact that the existing method seems to only account for single distortions. However, one can compose distortions (resize first, then blur later for instance). It is currently unclear to me as to how the method would work given these settings.
3. The limitations should be discussed with further detail. The issues that the current method has with respect to data coming from different sources would be interesting and insightful to the community.
4. The plots have extremely small text and it can be hard to follow, it would be better if the text in the plots are much bigger than they currently are. Especially for the plots in the appendix and Fig 2.

Minor Weaknesses
1. Line 42-43: This statement is not correct. A lot of detectors use common post-processing operations as part of their training [1,2].

References,
1. Wang, S. Y., Wang, O., Zhang, R., Owens, A., & Efros, A. A. (2020). CNN-generated images are surprisingly easy to spot... for now. In Proceedings of the IEEE/CVF conference on computer vision and pattern recognition (pp. 8695-8704).
2. Gragnaniello, D., Cozzolino, D., Marra, F., Poggi, G., & Verdoliva, L. (2021). Are GAN generated images easy to detect? A critical analysis of the state-of-the-art. arXiv preprint arXiv:2104.02617.

**Questions:**

1. Does the method currently account for multiple distortions, if not can it be made to account for this case?
2. For the training, why does equation 7 use MSE loss as opposed to the binary cross entropy loss?

---

> ### Author Response · Authors · 2025-11-20
> **Response to Reviewer DcA6 (Part 1)**
>
> 1. **Regarding Weaknesses (1):**  Thank you for the feedback. We will add detailed parameter specifications for all distortion types — including intensity ranges, application probabilities, and implementation settings — in Tables 1–2 of the revised manuscript. These details, previously only partially covered in Appendix D, are now fully documented. For evaluation consistency, during testing, we consolidated “Color Warming” and “Color Cooling” into a single category: “Color Shift”. See Tables 1–2 in the revised manuscript for full specifications.
>
>     Table 1: The types of distortions and their corresponding intensities used in the training set.
>     | Distortion Type   | Parameter Range  | Step        |
>     |-------------------|-------------------------------|-------------|
>     | JPEG              | [24, 96]                      |  8          |
>     | Gaussian blur     | [0.3, 3.0]                    | 0.3         |
>     | Noise             | [0.0002, 0.002]               | 0.0002      |
>     | Resize            | [0.20, 0.92]                  | 0.08        |
>     | Color warming     | [1.05, 1.5]                   | 0.05        |
>     | Color cooling     | [0.5, 0.95]                   | -0.05       |
>     | Brighten          | [0.05, 0.5]                   | 0.05        |
>     | Darken            | [-0.5, -0.05]                 | -0.05       |
>
>
>     Table 2: The types of distortions and their corresponding intensities applied in the test set.We only listed distortion types with tunable parameters. Distortions such as Histogram, Super-Resolution (2× upscaling), QQ, WeChat, Weibo, and Facebook involve no adjustable parameters and were therefore omitted from the table.
>     | Distortion Type   | Parameter Range  | Step        |
>     |-------------------|-------------------------------|-------------|
>     | JPEG              | [24, 100]                     |  2          |
>     | Gaussian blur     | [0.15, 5.0]                   | 0.15        |
>     | Noise             | [0.0001, 0.002]               | 0.0001      |
>     | Resize            | [0.5, 1.5]                    | 0.2         |
>     | Color shift       | [0.5, 1.5]                    | 0.2         |
>     | Saturation        | [1.9, 0.1]                    | 0.2         |
>     | Brighteness       | [-0.5, 0.5]                   | 0.2         |
>     | Sharpening        | radius: [1, 5], amount: [0.5, 3]                              | Random within ranges           |
>     | Mix1              | Random 2–4 from {JPEG, Blur, Noise, Resize}                 | Per-distortion random       |
>     | Mix2              | Random 1–3 from above set + JPEG                 | Per-distortion random       |
>
>     > **Note**: "Parameter Range" indicates the **closed interval** `[min, max]` for a single distortion. For `mix1` and `mix2`, parameter ranges are not fixed; instead, each component distortion randomly samples its intensity from its own range.
>     The "Sharpening" distortion uses the Unsharp Mask algorithm internally, controlled by radius (blur extent) and amount (enhancement strength).
>
>
> 2. **Regarding Weaknesses (2) and Questions (1):** Our method is not limited to single-distortion scenarios. Although during training each image is corrupted with only one distortion type (i.e., no mixed or sequential distortions are used), our model still demonstrates strong generalization to compositional distortions, where multiple distortions are applied sequentially — as evidenced by the performance on Mix1 and Mix2 in Table 2.
>
>     Specifically:
>
>     1. Mix1: Randomly selects 2–4 distortion types from {JPEG, Blur, Noise, Resize} and combines them randomly.
>
>     2. Mix2: Similarly selects 2–4 types, but fixes JPEG as the final distortion step (with others applied randomly before).
>
>     Crucially, despite no explicit training on such combinations, our Distortion Type Encoder learns to identify individual distortion signatures — enabling the model to implicitly decompose and respond to their combined effects. This indicates that our approach does not require modeling distortion sequences explicitly; instead, it leverages per-distortion knowledge to handle complex, real-world degradation patterns.
>
> 3. **Regarding Question (2):** The target “score” in our task is not a class probability but a continuous statistical metric derived from prior knowledge, which quantifies the detection performance of a data segment within a given interval. Since this score is computed as an empirical performance indicator — rather than representing the likelihood of a binary outcome — the problem is inherently a regression task. Accordingly, we use Mean Squared Error (MSE) loss to minimize the deviation between predicted and ground-truth performance scores. Binary Cross-Entropy (BCE), by contrast, assumes the output corresponds to a Bernoulli probability and is tailored for classification, making it mismatched to our objective. Hence, MSE is the natural and principled choice for optimization.

---

> ### Author Response · Authors · 2025-11-20
> **Response to Reviewer DcA6 (Part 2)**
>
> 4. **Regarding Weaknesses (3) on limitations:** We sincerely appreciate this valuable feedback. One key limitation of our current method is its reduced generalization to data from unseen domains — for instance, when applied to datasets collected under different conditions. This performance drop arises partly because our model incorporates detector-specific priors, which are effective within the training domain but may not transfer well under distribution shift.
>
>     We fully acknowledge this as a critical limitation. As future work, we aim to move beyond detector-specific priors and instead design models that directly learn the underlying relationship between input distortions and detection performance, without relying heavily on domain-specific assumptions. This shift would reduce sensitivity to data source variations by focusing on more intrinsic, transferable patterns of degradation. For example, rather than engineering features based on known statistics of a particular dataset, we will architectures that implicitly capture distortion-performance dynamics through end-to-end learning on diverse data sources. Such an approach could lead to more generalizable detectors that are less influenced by domain-specific artifacts during both training and inference. We believe this direction can help advance the development of reliable and deployable detection systems for heterogeneous, real-world environments.
>
> 5. Issues such as small text and plots that affect readability will be addressed in our subsequent revised version. Regarding the issue you raised about the phrasing in “Lines 42–43,” we have reviewed it and agree that the original wording was inappropriate. We will revise it to “Unfortunately, if a detector is trained exclusively on clean data, it remains largely unaware of distortions; consequently, as shown in Figure 1 (a), it reports only a binary decision (real or fake) without quantifying how much the decision should be trusted. Lacking this confidence signal, downstream systems (e.g., human fact-checkers, multi-detector pools) cannot assess which detector performs better on a certain sample, and thus are unable to reasonably abstain or select among detectors, raising both reliability and usability concerns. Even when distortion augmentations are included during training, this issue still persists.” Thank you for your helpful feedback.

---

### Official Review · Reviewer_iFaB · 2025-11-01

**Soundness:** 3
**Presentation:** 4
**Contribution:** 3
**Rating:** 8
**Confidence:** 4

**Summary:**

The authors propose to train a predictor for calibration-type confidence in the sense of probability of the prediction being wrong. They do this specifically for GAN-generated image detection.

To do this, they use detector features, features from a predictor of distortion type, features from a predictor of no-reference image quality, and use this with an MLP on top to predict the confidence via regression.

They perform experiments on correlation between the predicted and the true calibration, they show the usefulness of the predictor for top-1 routing to GAN-generated image detectors, and for confidence based vote abstention, evaluated by a ranking measure.

**Strengths:**

- They perform experiments beyond just training the calibrator.
    - It shows its usability for top-1 routing and for vote abstention / or low confidence flagging.
- if one would want to hide the fact that an image was generated by a deep learning model, then image distortions are a natural candidate for obfuscation, so the setting makes sense
- Clear idea
- well readable paper

**Weaknesses:**

- it has not the greatest novelty, it is not what one would think has to be shown as an oral
- The dataset used for the main experiments, PROGAN. is from the pre-diffusion model era.

It would be better to see the results for distortions also for diffusion datasets. They do this for the cross-evaluation in section 5.4 but it would be good to have done it also for section 5.3 and also for the confidence evaluation.

**Questions:**

none

---

> ### Author Response · Authors · 2025-11-20
> **Response to Reviewer iFaB**
>
> **Regarding the novelty:** We address the challenge that existing forensic models lack a reliable confidence estimation mechanism under distortion. Our approach models the relationship between the detector and distortions from an image-quality perspective, enabling the prediction of distortion-aware forensic scores. Therefore, we believe our problem definition, modeling perspective, and resulting framework provide meaningful contribution to the community.
>
> **Regarding the used datasets:** Thank you for this valuable suggestion. In response, we have extended both Section 5.3 (multi-detector fusion) and the confidence-based filtering evaluation to the SD14 dataset from GenImage [1], using the exact same experimental protocol as for PROGAN. During training, FreqNet and SAFE failed to converge under the required augmentation pipeline—likely due to sensitivity to the high-frequency artifacts and distributional characteristics of diffusion-generated images. Therefore, we report results only for CD, ND, NPR, and C2P. All models, including $DACOM_{DISTS}$ , are trained on SD14, and baselines use temperature-scaled logit calibration.
> The distortion types and data volume used during training and evaluation are fully consistent with the original paper's configuration. We compare our proposed(using $DACOM_{DISTS}$ as an example) method against the outputs of the calibrated detectors.
>
> As shown in the tables below, $DACOM_{DISTS}$ consistently outperforms baseline calibration methods across both seen and unseen distortions, delivering superior average and worst-case performance. It also enables more effective precision-coverage trade-offs in confidence-based filtering on SD14 and cross-dataset evaluations with SD15. These results demonstrate DACOM’s robustness on diffusion-era content and will be incorporated into the revised manuscript.
>
> Table 1: Multi-detector fusion performance on SD14 (%; Acc↑ / AP↑)
>
> | Method              | Seen distortion   | Unseen distortion          | Average          | Worst          |
> |---------------------|--------------|---------------|---------------|---------------|
> | Logit Calib.        | 91.85/99.00   | 94.64/98.67   | 93.59/98.79    | 72.70/90.14    |
> | $DACOM_{DISTS}$ | **97.70/99.70** | **96.10/98.95** | **96.70/99.23** | **81.60/93.38** |
>
>
>
> Table 2: Single-Distortion filtering (%) in evaluation dataset (Filt-Prop. means Filtering
> Proportion)
>
> | Method              | Distortion BAcc↑/EER↓   | Filt-Prop. 0.10          | Filt-Prop. 0.20          | Filt-Prop. 0.30          | Filt-Prop. 0.40          |
> |---------------------|--------------|---------------|---------------|---------------|---------------|
> | Logit Calib.        | 87.77/11.82  | 89.88/11.46   | 91.11/10.78   | 92.62/9.72    | 94.20/8.38    |
> | $DACOM_{DISTS}$ | 87.77/11.82  | **90.31/7.97** | **92.68/5.22** | **94.29/4.02** | **95.50/2.92** |
>
> Table 3: Single-Distortion filtering (%) in cross-dataset
>
> | Method              | Distortion BAcc↑/EER↓   | Filt-Prop. 0.10          | Filt-Prop. 0.20          | Filt-Prop. 0.30          | Filt-Prop. 0.40          |
> |---------------------|--------------|---------------|---------------|---------------|---------------|
> | Logit Calib.        | 87.74/11.60  | 89.78/11.31   | 91.18/10.81   | 92.62/9.93    | 94.06/8.60    |
> | $DACOM_{DISTS}$ | 87.74/11.60  | **90.28/7.77** | **92.69/5.18** | **94.34/3.89** | **95.55/2.95** |
>
> Table 4: Multi-Distortion filtering (%) in evaluation dataset
>
> | Method              | Distortion BAcc↑/EER↓   | Filt-Prop. 0.10          | Filt-Prop. 0.20          | Filt-Prop. 0.30          | Filt-Prop. 0.40          |
> |---------------------|--------------|---------------|---------------|---------------|---------------|
> | Logit Calib.        | 76.81/21.76  | 77.77/22.66   | 78.19/23.63   | 78.44/24.42    | 78.62/24.75    |
> | $DACOM_{DISTS}$ | 76.81/21.76  | **78.69/18.93** | **81.01/15.80** | **83.00/13.90** | **84.51/12.48** |
>
> Table 5: Multi-Distortion filtering (%) in cross-dataset
>
> | Method              | Distortion BAcc↑/EER↓   | Filt-Prop. 0.10          | Filt-Prop. 0.20          | Filt-Prop. 0.30          | Filt-Prop. 0.40          |
> |---------------------|--------------|---------------|---------------|---------------|---------------|
> | Logit Calib.        | 76.21/22.12  | 77.36/23.12   | 77.74/24.17   | 77.97/25.00    | 77.99/25.61    |
> | $DACOM_{DISTS}$ | 76.21/22.12  | **78.01/19.52** | **80.52/16.32** | **82.54/14.17** | **83.95/12.76** |
>
>
> [1]Zhu M, Chen H, Yan Q, et al. Genimage: A million-scale benchmark for detecting ai-generated image[J]. Advances in Neural Information Processing Systems, 2023, 36: 77771-77782.

---

> > ### Comment · Reviewer_iFaB · 2025-11-25
> >
> > The reviewer has read the rebuttal, the original score is already high and matching the manuscript quality.

---

> > > ### Author Response · Authors · 2025-11-28
> > > **Response to Reviewer iFaB**
> > >
> > > We appreciate you maintaining your score and for your valuable feedback, which we will incorporate to further improve our work.

---

### Author Response · Authors · 2025-12-03
**Summary of Rebuttal and Discussions**

We acknowledge the recent notice from the Program Chairs regarding the anonymity breach on OpenReview and **would like to express our sincere gratitude to the Area Chairs, Reviewers, and Program Committee for their professionalism and dedication under these exceptional circumstances.**

Below, we concisely highlight the key technical points from our rebuttal to ensure that the contributions and merits of our work are clearly conveyed.

---
### Summary of Key Responses to Reviewer Concerns

**1. Robustness to Training-Set Choice of DACOM (Reviewers iFaB & 5Ewd)**

- We replicated the main protocol on SD14 (diffusion) by training the detectors and DACOM under the same distortion setup. DACOM remains consistently better than logit calibration under both seen and unseen distortions, **indicating the framework is not tied to a single training benchmark**; the gains also carry over in SD14→SD15 evaluation.

**2. Design Rationale, Multi-Distortion Generalization, and Stated Limitations (Reviewers DcA6 & 5Ewd)**

- **Design Rationale:** FR-IQA scores alone are not distortion-discriminative—different distortion types can yield similar quality values yet trigger very different detector failures—so DACOM combines quality binning with distortion-type awareness.

- **Multi-Distortion Generalization:** Although trained with single distortions, DACOM generalizes to randomly compositional distortions without explicit sequence modeling, via learned distortion signatures.

- **Stated Limitations:** We explicitly acknowledge limitations: detector-specific priors can reduce transfer to unseen domains, and label generation incurs extra cost—both highlighted as future directions.

**3. Efficiency, Deployment Overhead, and Practical Considerations (Reviewers 5Ewd & 2NT5)**

- We report DACOM’s parameter count, FLOPs, inference latency, and its share of end-to-end inference time.

- We discuss deployment trade-offs of confidence-based abstention: it reduces coverage but serves as a standard safety mechanism in high-stakes settings, with added ethical discussion.

**4. Experimental Details and Presentation Quality (Reviewers DcA6 & 2NT5)**


- **We clarify key design/implementation choices and strengthen reporting:** motivate MSE for regressing performance-derived scores, add label-function ablations, and report F1/AUC-PR in addition to BAcc/EER.

- We provide complete distortion specifications (training/test ranges and step sizes) and clarify evaluation conventions.

- We improve readability and correct imprecise wording in the manuscript.
---
### Summary of Discussions

We sincerely thank the Area Chairs and Reviewers for their time and dedicated efforts throughout the review process.
During the discussion, we were encouraged by the reviewers’ positive assessments of our work: **Reviewer iFaB** highlighted its **“clear idea”** and **“well readable”** presentation; **Reviewer DcA6** described the problem as **“both interesting as well as practically relevant”**; and **Reviewer 5Ewd** considered it a **“fresh and meaningful contribution to the forensics community.”** Additionally, **Reviewer 2NT5** noted that our framework is **“novel”** and **“orthogonal and complementary to existing works.”**

In brief, DACOM estimates an image-level confidence score by combining full-reference image quality assessment (FR-IQA), intermediate forensic features from the detector, no-reference image quality assessment (NR-IQA), and distortion-type cues, and uses DACOM as a detector-side assistant to improve the overall accuracy of the detection system. During rebuttal, the reviewers raised concerns regarding (i) potential dependence on specific datasets, (ii) limitations of the experimental setup (e.g., composed distortions), (iii) computational overhead, and (iv) clarity and completeness of reporting. **We addressed these points through additional experiments, targeted ablations, and manuscript revisions.**


### Key Contributions of DACOM

- We introduce the **Distortion-Aware Confidence (DAC) score** and conduct an in-depth analysis revealing the correlation between FR-IQA scores and forensic accuracy. By leveraging DACOM to estimate DAC, **we achieve sample-level confidence assessment that assists the detector’s decision-making.**

- Compared with simple logit calibration, DACOM **improves performance in selective filtering** (better accuracy–coverage trade-offs) and **enhances multi-detector routing effectiveness.**

### Reviewer Feedback and Score Updates

Following our response, **Reviewer iFaB maintained their positive rating, noting that the original score appropriately reflected the paper’s quality**. **Reviewer 5Ewd, who previously stated they would be inclined to raise their score if concerns were addressed, increased their rating from 6 to 8.**

Finally, we thank the Area Chairs and Reviewers again for their constructive feedback, which has helped improve the paper. We hope this summary is helpful for the final assessment.

---

### Meta-Review · Area_Chair_hJou · 2025-12-30

**Summary:**

This paper received scores of 8,6,6,8.  Initial concerns include incremental novelty, outdated evaluation benchmark, some missing experimental details, lacking multiple composed distortions, clarity, concerns regarding the use of detector performance for labeling, limited discussion on robustness training, limited training data scale, metric choice, inference cost, and lacking risk/mitigation discussions.

**Reviewer Concerns:**

Most of the concerns were adequately addressed by the rebuttal.  Remaining concerns include latency costs, limited data diversity, and detector-specific labeling but these are clearly quantified and acknowledged in the rebuttal.

**Reviewer Scores:**

The reviewers would likely have remained positive, and reviewer 5Ewd may have increased their score to an 8.  This would have resulted in final scores of 8,6,8,8.

---

### Decision · Program_Chairs · 2026-01-26

Accept (Poster)